# Risk Assessment and Statistical Significance in the Age of Foundation Models

## Abstract

We propose a distributional framework for assessing socio-technical risks of foundation models with quantified statistical significance. Our approach hinges on a new statistical relative testing based on first and second order stochastic dominance of real random variables. We show that the second order statistics in this test are linked to mean-risk models commonly used in econometrics and mathematical finance to balance risk and utility when choosing between alternatives. Using this framework, we formally develop a risk-aware approach for foundation model selection given guardrails quantified by specified metrics. Inspired by portfolio optimization and selection theory in mathematical finance, we define a *metrics portfolio* for each model as a means to aggregate a collection of metrics, and perform model selection based on the stochastic dominance of these portfolios. The statistical significance of our tests is backed theoretically by an asymptotic analysis via central limit theorems instantiated in practice via a bootstrap variance estimate. We use our framework to compare various large language models regarding risks related to drifting from instructions and outputting toxic content.

## 1 Introduction

Foundation models such as large language models (LLMs) have shown remarkable capabilities redefining the field of artificial intelligence. At the same time, they present pressing and challenging socio-technical risks regarding the trustworthiness of their outputs and their alignment with human values and ethics (Bommasani et al., 2021). Evaluating LLMs is therefore a multi-dimensional problem, where those risks are assessed across diverse tasks and domains (Chang et al., 2023).

In order to quantify these risks, Liang et al. (2022); Wang et al. (2023); Huang et al. (2023) proposed benchmarks of automatic metrics for probing the trustworthiness of LLMs. These metrics include accuracy, robustness, fairness, toxicity of the outputs, etc. Human evaluation benchmarks can be even more nuanced, and are often employed when tasks surpass the scope of standard metrics. Notable benchmarks based on human and automatic evaluations include, among others, Chatbot Arena (Zheng et al., 2023), HELM (Bommasani et al., 2023), MosaicML's Eval, Open LLM Leaderboard (Wolf, 2023), and BIG-bench (Srivastava et al., 2022), each catering to specific evaluation areas such as chatbot performance, knowledge assessment, and domain-specific challenges. Traditional metrics, however, sometimes do not correlate well with human judgments. Aiming for a better alignment with human judgments, some approaches utilize ChatGPT/GPT-4 for natural language generation evaluations (Liu et al., 2023; Zhang et al., 2023; Hada et al., 2023).

A comprehensive evaluation of LLMs requires addressing the following critical considerations:

1. ***Interpretability.*** Evaluation of foundation models is multi-dimensional in nature and multiple metrics assess the models on different socio-technical dimensions that probe the trustworthiness of their outputs and their adherence to shared values and ethics. *It is critical to establish an aggregate-level measure to facilitate the interpretation and effective communication of the evaluation results.*

2. ***Risk Assessment.*** In natural language (and other) applications, metrics quantify important guardrails such as model's toxicity, safety, or robustness. Therefore, a comprehensive evaluation framework must incorporate a risk assessment that quantifies associated risks.

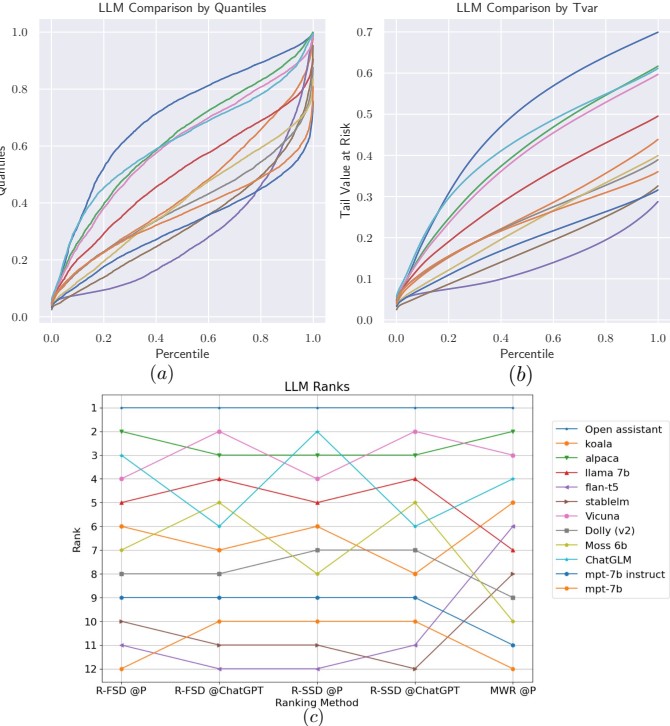

Figure 1: (a) Quantiles, (b) Tail Value at Risk (TVAR), of Metrics portfolio of an LLM, showing that TVAR (second-order stochastic dominance) more clearly ranks the models than the quantiles alone (first-order stochastic dominance). (c) Ranking of models using Relative First and Second Stochastic Dominance of Portfolios (R-FSD, R-SSD @P) versus ranking of models using Relative First and Second Stochastic Dominance of chatGPT evaluation scores and ranking by Mean Win Rate on the metrics portfolio. Note that (1) the metrics portfolio successfully approximates the chatGPT evaluation, since the @P rankings largely agree with the @chatGPT rankings; (2) the Relative Stochastic Dominance rankings outperform the baseline Mean Win Rate.

    This entails ranking models based on the assessment of failure modes and tail statistics[1], providing a nuanced understanding of potential pitfalls.

3. ***Statistical Significance.*** Evaluating machine learning models is intimately connected to statistical significance testing (SST), although this framework is still underutilized: Dror et al. (2018) reports almost 50% of ACL papers miss SST indicators. With the ever increasing parametric complexity of LLMs, obtaining a reliable SST in evaluating foundation models becomes ever more urgent.

We propose in this paper an evaluation framework that offers a principled solution and an efficient implementation that addresses each of these challenges. Our main contributions are:

1. ***Interpretable Metrics-Portfolio (Section 4).*** Drawing inspiration from econometrics and mathematical finance, we define a metrics-portfolio for aggregating metrics. This portfolio normalizes and aggregates metrics, yielding a single interpretable number assessing each output of a LLM. A higher value of the portfolio is preferable. We illustrate in Figure 1 panels (a) and (b) summary statistics of the metrics portfolio aggregating a total of $8$ automatic metrics computed using $5K$ samples from the Mix-instruct dataset (Jiang et al., 2023). In panel (c) we show that model ranking based on our metrics-portfolio aligns with human evaluation proxies such as chatGPT (Please refer to Appendix B for details of how chatGPT score is computed).

2. ***Risk Assessment via Second Order Stochastic Dominance (Section 2).*** Stochastic orders define partial orders on random variables and play a vital role in econometrics and mathe-

---

[1]I.e. understanding and quantifying low-probability high-risk events.

matical finance for comparing and selecting portfolios. We propose using stochastic order to select LLMs based on their metrics-portfolios. A portfolio dominates in the First Order Stochastic Dominance (FSD) if it has higher quantiles for all percentiles. However, in Figure 1 (Panel (a)), the quantiles of the metrics-portfolio of an LLM don't provide a clear ordering, and FSD doesn't adequately assess the risks of these models. Instead, we propose the use of Second Stochastic Dominance (SSD), where a portfolio dominates if it has higher Tail Values at Risk (TVAR) for all percentiles (also known as Conditional Value at Risk). TVAR, illustrated in Figure 1 (Panel (b)), represents normalized integrated quantiles, assessing the risks of low values in the portfolio. Small TVAR corresponds to fat left tails in the distribution of the portfolio, identifying risky LLMs as those with the lowest TVAR. For example, Flan-t5 emerges as the riskiest model in our running example.

3. ***Statistical Significance via Dominance Tests. (Section 3)*** Armed with these notions of stochastic dominance, we define statistics that assess the *relative* dominance of a model's portfolio on another (R-FSD and R-SSD in Panel (c) in Figure 1). We subject these statistics to an asymptotic analysis, proving central limit theorems that provide the foundation for hypothesis testing with false discovery rate control. We then perform stochastic dominance hypothesis testings between all pairs of models. Having adjusted the confidence level of these tests, we aggregate these pairwise rankings to a single rank via rank aggregation techniques such as the Borda Algorithm (de Borda, 1781). The resulting ranks, depicted in Panel (c) of Figure 1, highlight that the portfolio of automatic metrics (@P) leads to a similar ranking to chatGPT score (@chatGPT) for both first and second stochastic order. To underscore the importance of risk assessment, we present the ranking of the metrics-portfolio produced by the ubiquitous Min Win Rate (MWR) used in LLM benchmarks (Liang et al., 2022)(last column in Panel (c)). Flan-t5 ranks close to last with all other orders, but ranks 6 with MWR. This highlights that the ubiquituous MWR used in LLM benchmarks is risky for ranking LLMs as it does not take into account failure modes of the model, and we caution practitioners of its pitfalls.

## 2    STOCHASTIC DOMINANCE

We first review notions of stochastic dominance and their relation to downside risk measures and risk averse preference modeling. We use the notation of the seminal paper of Ogryczak & Ruszczynski (2002), and assume that the random variables are standardized so that larger outcomes are preferable. Throughout this Section, the reader can think of the random variable $X$ as a metric evaluating the performance of model $A$ on a specific test set. Likewise, $Y$ represents the evaluation of model $B$. We defer the definition of metrics portfolio to Section 4. In a multi-metric evaluation, as explained in the introduction, $X$ and $Y$ represent portfolios of evaluations of model $A$ and $B$ respectively.

### 2.1    FIRST AND SECOND ORDER DOMINANCE AND MEAN-RISK MODELS

**First Order Stochastic Dominance**    The First-order Stochastic Dominance (FSD) between real-valued random variables uses the right-continuous cumulative distribution (CDF) as a performance function. Specifically, for a real random variable $X$, define the first performance function $F_X^{(1)} : \mathbb{R} \to [0, 1]$ as the CDF: $F_X^{(1)}(\eta) = \mathbb{P}(X \leq \eta), \forall \eta \in \mathbb{R}$. The FSD of $X$ on $Y$ is defined as follows:

$$X \underset{\text{FSD}}{\succeq} Y \iff F_X^{(1)}(\eta) \leq F_Y^{(1)}(\eta), \forall \eta \in \mathbb{R}, \tag{1}$$

this intuitively means that for all outcomes $\eta$, the probability of observing smaller outcomes than $\eta$ is lower for $X$ than $Y$. An equivalent definition can be expressed using the quantile $F_X^{(-1)}$ (See e.g Ogryczak & Ruszczynski (2002)):

$$X \underset{\text{FSD}}{\succeq} Y \iff F_X^{(-1)}(p) \geq F_Y^{(-1)}(p), \forall p \in (0, 1], \tag{2}$$

where $F_X^{(-1)} : [0, 1] \to \overline{\mathbb{R}}$ is the left-continuous inverse of $F_X^{(1)}$: $F_X^{(-1)}(p) = \inf\{\eta : F_X^{(1)}(\eta) \geq p\}$ for $p \in (0, 1]$. We focus on this definition as it is more computationally and notationally friendly since the quantile function is always supported on $[0, 1]$.

**Second Order Stochastic Dominance** The Second-order Stochastic Dominance (SSD) is defined via the second performance function $F_X^{(2)} : \mathbb{R} \to [0, 1]$ that measures the area under the CDF: $F_X^{(2)}(\eta) = \int_{-\infty}^{\eta} F_X^{(1)}(x)dx$, for $x \in \mathbb{R}$, yielding:

$$X \underset{\text{SSD}}{\succcurlyeq} Y \iff F_X^{(2)}(\eta) \leq F_Y^{(2)}(\eta), \forall \eta \in \mathbb{R}. \tag{3}$$

Note that FSD implies SSD, hence SSD is a finer notion of dominance. While FSD implies that $X$ is preferred to $Y$ by any utility-maximizing agent preferring larger outcomes[2], Ogryczak & Ruszczyn-ski (2002) showed that SSD implies that $X$ is preferred to $Y$ by any *risk-averse* agent preferring larger outcomes.[3] Similarly to FSD, SSD can be measured with quantile functions via introducing the second quantile function also known as *integrated quantiles* $F_X^{(-2)} : (0, 1] \to \overline{\mathbb{R}}$

$$F_X^{(-2)}(p) = \int_0^p F_X^{(-1)}(t)dt, \text{ for } t \in (0, 1]. \tag{4}$$

Similarly to the FSD case, an equivalent more computationally friendly definition can be expressed in terms of the second quantile function (a proof of this equivalence can be found in Theorem 3.2 in Ogryczak & Ruszczynski (2002)):

$$X \underset{\text{SSD}}{\succcurlyeq} Y \iff F_X^{(-2)}(p) \geq F_Y^{(-2)}(p), \forall p \in (0, 1]. \tag{5}$$

This equivalence is not straightforward and is due to Fenchel duality between $F^{(2)}$ and $F^{(-2)}$. Using $p = 1$ we see that SSD implies $\mu_X \geq \mu_Y$, where $\mu_X$ and $\mu_Y$ are means of $X$ and $Y$.

**Mean – Risk Models (MRM)** As noted earlier SSD is linked to risk assessment via the second performance function $F^{(2)}(.)$ measuring expected shortfall, and the negative second quantile function $-F^{(-2)}(p)$ that is an assessment of expected losses given outcomes lower than the $p$-quantile.

**Definition 1** (Mean – Risk Models). *A mean – risk model of a random variable $X$ consists of the pair $(\mu_X, r_X)$, where $\mu_X$ is the mean of $X$, and $r_X$ is a functional that measures the risk of the random outcome $X$.*

The consistency of a mean – risk model with SSD is defined as follows:

**Definition 2** (SSD consistency of Mean – Risk Models). *A mean – risk model $(\mu_X, r_X)$ is $\alpha$−consistent with SSD, if for $\alpha > 0$ the following is true:*

$$X \underset{\text{SSD}}{\succcurlyeq} Y \implies \mu_X - \alpha r_x \geq \mu_Y - \alpha r_Y. \tag{6}$$

The ubiquitous mean – risk model in machine learning is $(\mu_X, \sigma_X)$, where $\sigma_X$ is the standard deviation. Unfortunately this model is not consistent with the SSD and has several limitations as it implies Gaussianity of the outcomes or a quadratic utility function. We give in Table 1 (Appendix F.1 ) risk measurements and their $\alpha$−consistency (proofs in Ogryczak & Ruszczynski (2002)).

## 2.2 RELAXATIONS OF STOCHASTIC DOMINANCE

Recalling the definitions of FSD and SSD in Equations (2) and (5), in the finite-sample regime it is hard to test for these relations as one needs to show the infinite-sample quantile or second quantile properties hold uniformly over all $p \in (0, 1]$. This difficulty motivated the relaxation of stochastic dominance to an almost stochastic dominance pioneered by Leshno & Levy (2002). These relaxations were revisited for the first order by Alvarez-Esteban et al. (2014) who later proposed an optimal transportation approach to assess almost first stochastic order (Del Barrio et al., 2018).

**Almost FSD ($\varepsilon$-FSD)** Following Leshno & Levy (2002), Del Barrio et al. (2018) relaxed FSD (Equation (2)) via the violation ratio of FSD:

$$X \underset{\varepsilon-\text{FSD}}{\succcurlyeq} Y \iff \varepsilon_{\mathsf{W}_2}(F_X, F_Y) = \frac{\int_0^1 (F_Y^{(-1)}(t) - F_X^{(-1)}(t))_+^2 dt}{\mathsf{W}_2^2(F_X, F_Y)} \leq \varepsilon, \tag{7}$$

---

[2]I.e. having an increasing utility function.

[3]I.e. having an increasing and *concave* utility function.

where $W_2$ is the Wasserstein -2 distance between $F_X$ and $F_Y$. This ratio corresponds to a measure of the "area" of violation of the FSD dominance of $X$ on $Y$. Note that $0 \leq \varepsilon_{W_2}(F_X, F_Y) \leq 1$, with value 0 if $X \underset{\text{FSD}}{\succ} Y$ and 1 if $Y \underset{\text{FSD}}{\succ} X$. For $\varepsilon \in (0, \frac{1}{2}]$, Figure 5a in Appendix G illustrates $\varepsilon$-FSD, dashed areas represent the violation set.

**Almost SSD ($\varepsilon$-SSD)** We define $\varepsilon$-SSD, for $\varepsilon \in (0, \frac{1}{2})$, by relaxing Equation (5) as follows:

$$X \underset{\varepsilon-\text{SSD}}{\succcurlyeq} Y \iff \varepsilon_{IQ}(F_X, F_Y) = \frac{\int_0^1 (F_Y^{(-2)}(t) - F_X^{(-2)}(t))_+^2 \, dt}{d_{IQ}^2(F_X, F_Y)} \leq \varepsilon, \qquad (8)$$

where $d_{IQ}$ is the $L_2$ distance between the Integrated Quantiles ($F^{(-2)}$). This ratio corresponds to a measure of the "area" of violation of the SSD dominance of $X$ on $Y$. Figure 5b in Appendix G illustrates the second order, dashed areas represent the violation set of SSD of $X$ on $Y$. Appendix D gives a more detailed account on almost stochastic dominance.

## 2.3 RELATIVE STOCHASTIC DOMINANCE

In the remainder of the paper, we refer to the FSD violation ratio as $\varepsilon_{W_2}(F_X, F_Y) \equiv \varepsilon^{(1)}(F_X, F_Y)$ and to the SSD violation ratio as $\varepsilon_{IQ}(F_X, F_Y) \equiv \varepsilon^{(2)}(F_X, F_Y)$. One of the shortcomings of almost stochastic dominance is the need to fix a threshold $\varepsilon$ on the violation ratio. When comparing two random variables, setting a threshold is a viable option. Nevertheless, when one needs to rank multiple variables $X_1, \ldots, X_k$ (considering all pairwise comparisons), setting a single threshold that would lead to a consistent relative stochastic dominance among the $k$ variables becomes challenging. To alleviate this issue, we draw inspiration from relative similarity and dependence tests (Bounliphone et al., 2016a;b) that circumvent the need for a threshold via relative pairwise testings.

For $\ell \in \{1, 2\}$ (i.e for FSD or SSD) we consider all pairs of violations ratios:

$$\varepsilon_{ij}^{(\ell)} = \varepsilon^{(\ell)}(F_{X_i}, F_{X_j}) \text{ for } i, j \in \{1 \ldots k\}, i \neq j,$$

noting that $\varepsilon_{ij}^{(\ell)} + \varepsilon_{ji}^{(\ell)} = 1$. Let $F = (F_{X_1}, \ldots F_{X_k})$. We define the one-versus-all violation ratio of the dominance of $X_i$ on all other variables $X_j, j \neq i$ :

$$\varepsilon_i^{(\ell)}(F) = \frac{1}{k-1} \sum_{j \neq i} \varepsilon_{ij}^{(\ell)}.$$

We then define relative stochastic dominance for both orders, R-FSD an R-SSD respectively:

$$X_{i_1} \underset{R-\text{FSD}}{\succcurlyeq} X_{i_2} \ldots \underset{R-\text{FSD}}{\succcurlyeq} X_{i_k} \iff \varepsilon_{i_1}^{(1)}(F) \leq \cdots \leq \varepsilon_{i_k}^{(1)}(F) \qquad (9)$$

$$X_{i_1} \underset{R-\text{SSD}}{\succcurlyeq} X_{i_2} \ldots \underset{R-\text{SSD}}{\succcurlyeq} X_{i_k} \iff \varepsilon_{i_1}^{(2)}(F) \leq \cdots \leq \varepsilon_{i_k}^{(2)}(F) \qquad (10)$$

In this definition of relative stochastic dominance, the most dominating model is the one with the lowest one-versus-all violation ratio and to test for relative dominance of $X_i$ on $X_j$ we can look at the following statistics:

$$\Delta\varepsilon_{ij}^{(\ell)}(F) = \varepsilon_i^{(\ell)}(F) - \varepsilon_j^{(\ell)}(F), \qquad (11)$$

and we have the following threshold-free test for relative order:[4]

$$X_i \underset{R-\text{FSD}}{\succcurlyeq} X_j \iff \Delta\varepsilon_{ij}^{(1)}(F) \leq 0 \qquad (12)$$

$$X_i \underset{R-\text{SSD}}{\succcurlyeq} X_j \iff \Delta\varepsilon_{ij}^{(2)}(F) \leq 0 \qquad (13)$$

---

[4]For comparing $k = 2$ random variables, these $r$-FSD and $r$-SSD tests reduce to 0.5-FSD and 0.5-SSD absolute tests, respectively.

## 3 TESTING FOR ALMOST AND RELATIVE STOCHASTIC DOMINANCE

Given empirical samples from $F_X$ and $F_Y$ we perform statistical testing of the almost and relative stochastic dominance of $X$ on $Y$ given empirical estimates of the statistics given in Sections 2.2 and 2.3. A key ingredient for quantifying the statistical significance of such tests is a central limit theorem that guarantees that the centered empirical statistics is asymptotically Gaussian at the limit of infinite sample size. Given $n$ samples from $F_X$ ($m$ from $F_Y$ respectively), we denote $F_X^n$ and $F_Y^m$ the corresponding empirical distributions. For $\varepsilon_0-$ FSD, Del Barrio et al. (2018) studied the following hypothesis testing $H_0 : X \underset{\varepsilon_0-\text{SSD}}{\not\succcurlyeq} Y$ versus the alternative $H_a : X \underset{\varepsilon_0-\text{SSD}}{\succcurlyeq} Y$. Using (2), this amounts to the following null hypothesis : $H_0 : \varepsilon_{\text{W}_2}(F_X^n, F_Y^m) > \varepsilon_0$. Del Barrio et al. (2018) showed the asymptotic normality of the empirical statistics:

$$\sqrt{\frac{mn}{m+n}} \left(\varepsilon_{\text{W}_2}(F_X^n, F_Y^m) - \varepsilon_{\text{W}_2}(F_X, F_Y)\right) \to \mathcal{N}(0, \sigma^2(F_X, F_Y)).$$

Del Barrio et al. (2018); Ulmer et al. (2022) propose to reject $H_0$ with a confidence level $1 - \alpha$ if:

$$\varepsilon_{\text{W}_2}(F_X^n, F_Y^m) \leq \varepsilon_0 + \sqrt{\frac{m+n}{mn}} \sigma^2(F_X, F_Y) \Phi^{-1}(\alpha), \tag{14}$$

where $\Phi^{-1}$ is the quantile function of a standard normal.

For the tests we propose below, we assume the following structure on the underlying CDFs to derive the corresponding central limit theorems (CLTs).

> **Assumption 1.** *[Regularity] Let the CDF $F$ be supported on the interval $[-M, M]$ for some constant $M$, and have pdf $f$ such that $\frac{f'(p)}{f^3(p)}$ is bounded for almost every $p$ for which $f(p) > 0$ (i.e. all $p$ in the support of $f$).*

$\varepsilon$**-SSD Testing** Similar to $\varepsilon$-FSD, using the definition in (5) we propose to test using the following null hypothesis for testing for $\varepsilon_0$-SSD:

$$H_0 : \varepsilon_{IQ}(F_X^n, F_Y^m) > \varepsilon_0$$

Supposing Assumption 1 holds for $F_X, F_Y$ and assuming $\frac{n}{n+m} \to \lambda$ for some $\lambda$, we state a Central Limit Theorem for the second order statistics in Appendix H (Theorem 1, proved in Appendix J.1). Similarly to (14), Theorem 1 suggests to reject $H_0$ with a confidence $1 - \alpha$ if :

$$\varepsilon_{IQ}(F_X^n, F_Y^m) \leq \varepsilon_0 + \sqrt{\frac{m+n}{mn}} \sigma_\lambda^2(F_X, F_Y) \Phi^{-1}(\alpha), \tag{15}$$

where (for the same reasons as the FSD case) $\sigma_\lambda^2$ is given by the central limit theorem.

**Relative Stochastic Dominance Testing** We turn now to relative stochastic dominance that we introduced in (12) and (13) for first and second orders. Given $n$ samples from $k$ random variables $(X_1 \ldots X_k)$, let $F = (F_1, \ldots, F_k)$ be the marginals of $X_i$ and $F_n = (F_{1n}, \ldots, F_{kn})$ denote the empirical marginals. To test for R-FSD (resp R-SSD) of $X_{i_1}$ on $X_{i_2}$ we propose to test the following null hypothesis:

$$H_0 : \Delta \varepsilon_{ij}^{(\ell)}(F_n) > 0, \ell = 1 \text{ or } 2$$

Assuming that each $F_i$ satisfies Assumption 1, we state in Appendix H a central limit theorem for the relative second order statistics (Theorem 2 proved in in Appendix J.2). A similar result holds for the relative first order statistics that we omit for brevity. Theorem 2 suggests to reject $H_0$ with a confidence $1 - \alpha$ if:

$$\Delta\varepsilon_{i_1,i_2}^{(2)}(F_n) \leq \sqrt{\frac{1}{n}}\sigma_{relative}^2(F_X, F_Y)\Phi^{-1}(\alpha) \tag{16}$$

where $\sigma_{relative}^2(F_X, F_Y)$ is given by the central limit theorem (similar test exists for R-FSD).

**Bootstrapping Heuristic** While the CLT above provides an asymptotic value for the variance, in practice (as in the ASO framework of (Ulmer et al., 2022)) we estimate the variance with a bootstrapping heuristic (Efron & Tibshirani, 1993). This estimate is nonasymptotic and hence should often be more accurate than the asymptotic value. Proving the consistency of the bootstrap for functions of quantiles is generally nontrivial (Shao & Tu, 2012), but recall that the stochastic ordering can be defined in terms of either quantiles or CDFs. In Appendix K we provide a bootstrap consistency proof for the absolute statistics based on the CDF, leaving the quantile based proof for future work.

**Multi-Testing Algorithm** Algorithm 1 given in Appendix C summarizes the multi-testing setup for both relative and almost (absolute) FSD and SSD. The main idea behind Algorithm 1 is to turn multi-testing to pairwise testings i.e testing for stochastic dominance between all pairs of models using relative (or absolute) FSD or SSD. In order to ensure that this multi-testing has a confidence level $1 - \alpha$, we correct the individual test's confidence level by dividing $\alpha$ by the number of all pairs (Bonferroni, 1936). Then in order to combine the pairwise rankings to a single rank, we use a simple Borda count (de Borda, 1781) rank aggregation algorithm.

## 4    DISTRIBUTIONAL RISK ASSESSMENT OF FOUNDATION MODELS

**Setup** In this section we consider the multi-metric evaluation setup of a foundation model $A : \mathcal{X} \to \mathcal{O}$, using $N$ metrics $m_i : \mathcal{O} \to \mathbb{R}, i = 1 \ldots N$, where $m_i$ are real valued functions evaluated on a test set $D$. Without loss of generality, assume that each of the metrics are standardized such that higher values of $m_i$ correspond to more desirable model performance. We model observed values for each metric $m_i$ as a continuous random variable $M_i$ with unknown CDF $F_{M_i}$. For a model $A : \mathcal{X} \to \mathcal{O}$ and a data sample $X \sim D$, we describe the evaluation of model $A$ with $m_i$ with the following random variable $M_i$: $M_i|A, X := m_i(A(X)), X \sim D, i = 1 \ldots N$, where the randomness arises from the data sampling procedure $X \sim D$, and (if applicable) the stochasticity of the model $A$, for example if the model uses sampling to compute its output.

**Metrics Portfolio Selection using Stochastic Dominance** Let $\lambda = (\lambda_1, \ldots, \lambda_N)$ be a probability vector that represents the importance of the $m_i$ metrics to the model's end user. Inspired by the portfolio optimization literature, we model the user return from a model as a *portfolio of metrics $m_i$ evaluated on a test set $D$*. Following (Ulan et al., 2021; Belgodere et al., 2023), we define this portfolio as an Archimedean copula, which forms a weighted geometric mean of the CDFs:

$$R_A(X) = \exp\left(\sum_{i=1}^{N}\lambda_i \log F_{M_i}(m_i(A(X)))\right) = \prod_{i=1}^{N} F_{M_i}^{\lambda_i}(m_i(A(X))). \tag{17}$$

Note that (17) normalizes the metrics using the CDF of the metric $M_i$, eliminating the issue of differing dynamic ranges. This CDF should be formed by pooling together the evaluations on all samples and from all models being compared, to ensure that the various $R_A$ are comparable. The CDF normalization is monotonic and hence it preserves the order of each metrics and allow us to aggregate in the probability space the metrics using a simple weighted geometric mean. Computing $R_A(X)$ for all test samples $X$, we can therefore characterize the distribution of the metric portfolio of the model $A$. To compare two models it is enough to compare their corresponding portfolios, specifically, Model $A$ is preferred to Model B using $\varepsilon$- or R-SSD:

$$R_A(X) \underset{\varepsilon- \text{ or } R-\text{SSD}}{\succcurlyeq} R_B(X). \tag{18}$$

Similar tests can be performed for FSD.

**Multiple Models Comparison** Given $k$ models $A_\ell, \ell = 1 \ldots k$ and their evaluations $m_i(A_\ell(X)), X \sim D, i = 1 \ldots N$, we pool all model evaluations for a metric to estimate the

CDF of each metric $F_{M_i}$ and construct a portfolio for each model $R_{A_\ell}(X)$. We use our Relative Stochastic Dominance testing introduced in Section 3 and in Algorithm 1 to rank the models by their metrics portfolio in relative SSD or FSD with a confidence level $1 - \alpha$.

**Per Metric Stochastic Dominance and Rank Aggregation** We also explore another approach for multi-testing, by considering the stochastic dominance of the models on per-metric basis. This amounts to computing $N$ relative stochastic orders for each $\mathcal{M}_i = (m_i(A_1(X)), \ldots, m_i(A_\ell(X)))$, $i = 1 \ldots N$. This amounts to producing via Algorithm 1 a relative ranking $\pi_i$ of the models based on $\mathcal{M}_i$. A single rank $\pi$ is then obtained via rank aggregation with uniform weighting on the per-metric rankings $\pi_i, i = 1 \ldots N$. We use for rank aggregation the R package of (Pihur et al., 2009). For more details on rank aggregation, the reader is referred to Appendix F.3.

# 5 EXPERIMENTS

## 5.1 VALIDATION OF STATISTICAL SIGNIFICANCE

We examine the statistical properties of our tests as a function of sample size. We purposely design synthetic score distributions to represent challenging problems comprising large overlap between the distributions and considerable violation ratio, but where one would still like to have an ordering among the variables. For this we consider the two Gaussian variables $X \sim \mathcal{N}(0, 1)$ and and $Y \sim \mathcal{N}(0.5, 2)$. Figure 6 in Appendix L.1 shows that our tests have desirable statistical properties. We also perform synthetic experiment on fat tailed distribution such as log normal (Figure 7 App. L.1).

## 5.2 LLM EVALUATION WITH STOCHASTIC DOMINANCE

We showcase LLM evaluation with stochastic dominance to assess two risks: drifting from instructions and outputting toxic content. The following datasets correspond to each risk we assess.

**Mix-Instruct Evaluation Data** We use the data from (Jiang et al., 2023), that consists of an instruction, an input sentence and an expected output from the user, as well as the output of a set of different LLMs. The dataset consists of a training set of 100K samples and a test set of 5K samples. (Jiang et al., 2023) used automatic metrics such as BARTscore and BLEU score comparing the LLM generation to the expected output in order to evaluate if each LLM followed the instruction. (Jiang et al., 2023) used also chatGPT to evaluate the generations. The number of automatic metrics $N$ is 8, the total number of evaluated models $k$ is 12. Metrics are unified so that larger values are preferred.

**Toxicity Evaluation** We use the real toxicity prompts dataset of Gehman et al. (2020), and generate prompts completions from the Llama 2 7b , Llama 2 13b, Llama 2 70b , MosaicML MPT 30b and Tiiuae Falcon 40b models available in Opensource ($k = 5$ models). We select two sets of prompts: toxic prompts (toxicity $> 0.8$, that gives ~10K prompts ) and non-toxic prompts (toxicity $< 0.2$, from which we randomly sample 10K). We sample from each model, 10 completions per prompt using nucleus sampling (top-$p$ sampling with $p = 0.9$ and a temperature of 1). This procedure yields a dataset of ~200K sentence completions per model. We evaluate the toxicity of these generations using the Perspective API, on the following toxicity metrics ($N = 6$ metrics): Toxicity, Severe toxicity, Identity Attack, Insult, Profanity and Threat. Following Liang et al. (2022), we evaluate the toxicity of generated completions only and refer to this as **_Gen Only_** evaluation. In order to also give the context of the completion, we prepend the model generation with the prompt and evaluate the full sentence using Perspective API. We refer to this as **_Prompt+Gen_**. The polarity of all toxicity metrics is unified so that high values refer to non toxic content (we use $-\log$ probabilities of Perspective API outputs).

**Evaluation Protocol and Baselines** We evaluate each of the use cases (instruction following and toxicity) using the following absolute stochastic dominance tests: (1) $\varepsilon$-FSD (corresponds to the ASO evaluation of Ulmer et al. (2022)) for $\varepsilon = 0.08, 0, 25, 0.4$. (2) our proposed $\varepsilon$-SSD using the same values for $\varepsilon$, (3) our relative stochastic dominance R-FSD and R-SSD tests, (4) the Mean $-$ Risk models described in Table 1, and (5) the ranking produced by the Mean Win Rate (**MWR**) used by LLM leaderboards such as HELM (Liang et al., 2022). As noted in Section 4, we either perform these tests on a *metrics portfolio* (given in Equation (17)) – we refer to this as **test @ P**; or on a per metric basis leading to $N$ rankings of the models that we reduce to a single ranking via Rank

Aggregation (RA) (Pihur et al., 2009) – we refer to this as **RA(test @ M)**. In this naming convention, **test** takes values in $\{\text{MWR}, \varepsilon\text{-FSD}, \varepsilon\text{-SSD}, \text{R-FSD}, \text{R-SSD}, \text{Mean} - \text{Risk Model } (\mu_X - r_X)\}$ where $r_X$ is a chosen risk from Table 1. We perform all our statistical tests with a significance level $\alpha = 0.05$, and use 1000 bootstrap iterations.

**Efficient Implementation** We compare the computational complexity of our implementation for computing all stochastic orders to that of the `Deep-Significance` package (deepsig, 2022) which implements $\varepsilon$-FSD in the ASO framework (Ulmer et al., 2022), on the task of comparing models on the Mix-Instruct dataset (sample size 5K, $k = 12$ models). Using the `Deep-Significance` implementation of MULTI-ASO in (Ulmer et al., 2022) for $\varepsilon = 0.25$ with just 3 bootstrap iterations[5], the test completes in 15min50s (averaged over 7 runs). Our code for relative and absolute testing performs all tests at once and relies on caching vectorization and multi-threading of the operations. Our code completes all tests in an average of just 17.7 s with 1000 bootstraps. Experiments were run on a CPU machine with 128 AMD cores, of which 2 were used.

**Mix-Instruct Results and Analysis** Figure 1 and Table 2 in Appendix and summarize the rankings we obtain for different models using the different tests described above. Note that we compare here our portfolio approach versus a ChatGPT score evaluation of the models (See Appendix B for ChatGPT evaluation). We see that our portfolio agrees with this human evalutaion proxies on both R-SSD and R-FSD orders. On the other hand, as we show in Appendix A, our portfolio approach also agrees with per metric aggregation for both R-FSD and R-SSD while being 7x faster. When compared to the Mean Win Rate currently used in LLM leaderboards such as HELM (Liang et al., 2022), we see that it leads to different orderings than FSD and SSD. For example the flan-t5 model is ranked 5 or 6 by MWR with rank aggregation and portfolio respectively. In contrast, for R-FSD and R-SSD it is given a low ranking $(8, 11)$ or $12$. This is due to the fact that MWR only counts wins and does not take into account how fat is the left tail of the distribution of the metric being assessed, possibly leading to overevaluation of risky models. When comparing R-FSD and R-SSD to each other, we see some changes in the ranking in near or adjacent positions. Remarkably, the R-SSD ordering agrees with the rank aggregation of all (consistent) mean – risk models, confirming the theoretical link between second order dominance and risk averse decision making. Nevertheless, as shown in Appendix A R-SSD has lower variance in smaller sample regime. Table 3 in Appendix L shows that R-FSD and R-SSD are consistent with $\varepsilon$-FSD and SSD respectively for various values of $\varepsilon$ on this dataset. While it is common to give radar plots of MWR for a metric or an average of the metric, we give in L a radar plot (Figure 8) for each of the Mean – Risk models, to aid practitioners in visualizing and selecting models in a risk aware manner.

**Toxicity Results and Analysis** Table 4 in Appendix L summarizes the results of our tests. We make a few observations: First, overall the portfolio approach agrees well with the rank aggregation of per-metric rankings. The portfolio is more computationally efficient as it needs to run the stochastic dominance test only on the portfolio, rather than running $N$ tests and aggregating them via rank aggregation. Secondly, on this dataset the R-FSD and R-SSD agree, with a few exceptions. Interestingly, when comparing models on model generation only, on toxic prompts MosaicML MPT stands out, while on non toxic prompts Llama2 7B stands out and on the combined set Mosaic ML MPT stands out. When evaluating the toxicity of the context (Prompt + Gen), Llama70B stands out on toxic prompts, Llama7b stands out on non toxic prompts and MosaicML MPT still stands out on the combined set. This study shows that the evaluation problem is not only challenging in terms of the statistical significance of the test, but also with regards to the conditioning on which data the evaluation is performed. The stability of the ranking across all methods, on the combined set suggests that rank stability can be a criterion to assess the representativity of the evaluation set.

## 6 CONCLUSION

In this paper we introduced a distributional framework for risk assessment and comparison of foundation models based on multi-metric evaluations. Our framework is of interest beyond the current applications presented here by providing statistical significance while ranking assets for decision making. We believe our tools for training models to be risk averse can be of significant use to practitioners and serve as a stepping stone towards solving the AI alignment problem.

---

[5]Limited to 3 for computational reasons.

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

CONTENTS

## A  ABLATION STUDIES

**Metrics Aggregation Versus Portfolio**    For portoflio, computing ranking using FSD and SSD including the portfolio computation on $5K$ samples for 5 bootstrap samples , we have mean execution time of $32.01 \pm 4.51$ s.  For FSd and SSD ranking computation for all metrics, followed by rank using pearson distance the execution time is of $254.99 \pm 16.76$ s. On the other hand, we observe on the mix-instruct dataset a consistency of ranks between these two approaches (FSD or SDD on port-

folio & FSD or SSD on all metrics followed by rank aggregation) as quantified by the kendall-tau similarity between the ranks:

1. Kendall Tau(R-SSD@P, RA(R-SSD@M)) = 0.848
2. Kendall Tau(R-FSD@P, RA(R-FSD@M)) =0.878

We see that these two approaches lead to similar ranks while portfolio approach leads to 7x speedups.

**Portfolio versus Mean Risk Models and MWR**

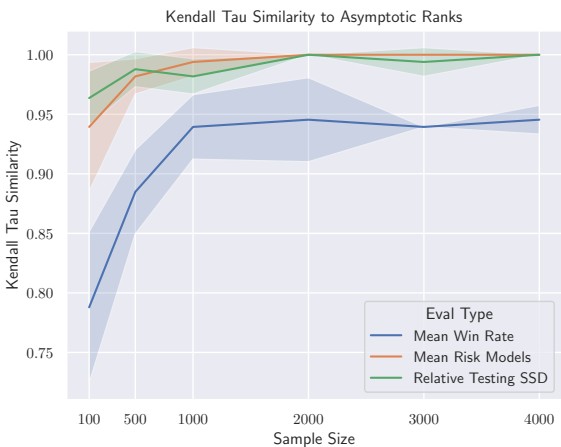

Figure 2: We compute ranking resulting from each ranking method using varying sample sizes from 100 to 5K. We repeat each experiment 5 times. We report for each method, the kendall-tau similarity between resulting ranks at each sample to the asymptotic rank at $5K$. We see that Relative SSD on portfolio is more stable in sample size than rank aggregation of all Mean Risk Models and than MWR on portfolio. Furthermore Relative SSD on portfolio has lower variance than aggregation of Mean Risk Models on portfolio.

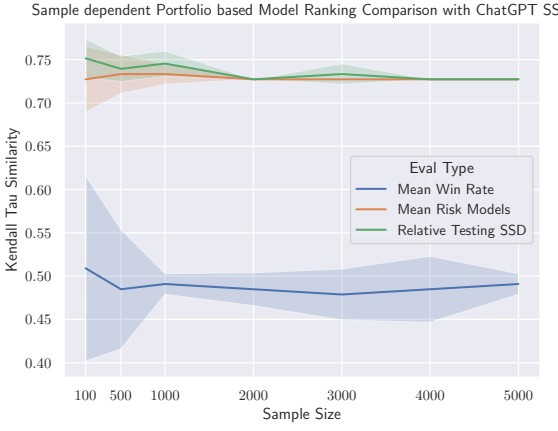

Figure 3: We use the same setup of the Figure above, but instead of kendall tau distance to asymptotic rank of each method. We plot the distance to R-SSD @ChatGPT rank. We see that MWR is inconsitent with rank while both R-SSD @P and RA(MRM @P) have a kendall-tau similarity between 0.7-0.75, with a lower variance for R-SSD in small sample regime.

## B  TRANSFORMING DISCRETE RELATIVE CHATGPT SCORES TO ABSOLUTE REAL VALUED SCORES

We follow Jiang et al. (2023) in mapping discrete chatGPT scores to real valued ones. Note that chatGPT scores for comparing models A and B are discrete and are one of these 4 options: A is better, B is better, Both are equally good, Both are equally bad.

Given $m$ models we construct for each prompt sample $\ell = 1 \ldots N$ a $m \times m$ comparison matrix with chatGPT:

$$X_{\ell,ij} = +1, X_{\ell,ji} = -1 \text{ if model } i \text{ is better}$$

$$X_{\ell,ij} = -1, X_{\ell,ji} = +1 \text{ if model } j \text{ is better}$$

$$X_{\ell,ij} = X_{\ell,ji} = +0.5 \text{ if model } i \text{ and } j \text{ equally good}$$

$$X_{\ell,ij} = X_{\ell,ji} = -0.5 \text{ if model } i \text{ and } j \text{ equally bad}$$

Then each model will define the following scalar score at each sample $\ell$:

$$s_{\ell,i} := \sum_{j=1}^{m} (X_{\ell,ij} - X_{\ell,ji}).$$

hence we have a distribution of chatGPT score for each model :

$$p_i = \frac{1}{N} \sum_{i=1}^{N} \delta_{s_{\ell,i}}, i = 1 \ldots m.$$

Note that the scores $s_{\ell,i}$ take on even integer values between $-2m$ and $2m$ inclusive, we treat the support as continuous and consider the following kernel density estimator with Gaussian kernel of width $\sigma$:

$$\hat{p}_i^{(\sigma)}(t) = \frac{1}{N} \sum_{i=1}^{N} \varphi \left( \frac{t - s_{\ell,i}}{\sigma} \right), t \in \mathbb{R}, i = 1 \ldots m,$$

where $\varphi$ is the standard normal density. In Figure 4 below we plot chatGPT scores kernel density estimates for two models, openassistant and flan-t5:

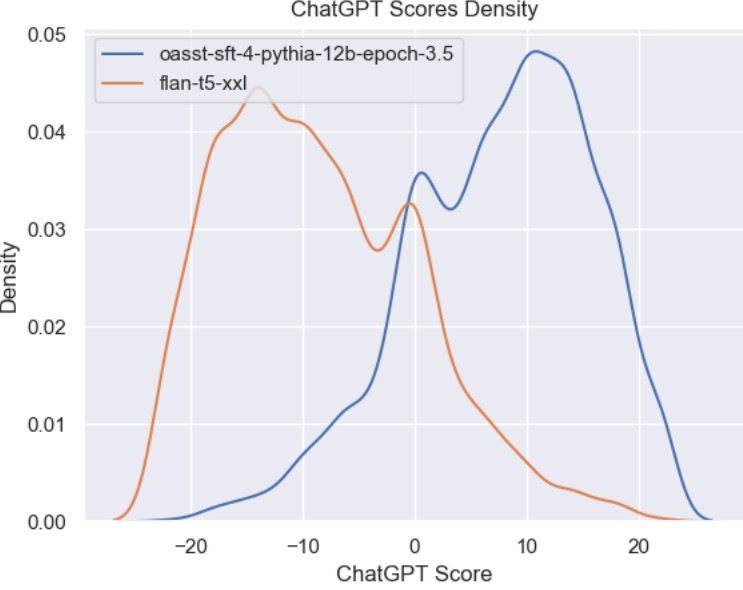

Figure 4: ChatGPT density scores for two models, open-assistant has clearly higher scores than the Flan-t5 models

## C  MULTI-TESTING ALGORITHM FOR RELATIVE AND ALMOST STOCHASTIC DOMINANCE

Our multi-testing algorithm for relative and almost stochastic dominance is detailed in Algorithm 1.

In a nutshell our multi-testing consists of the following steps:

1. For evaluation of each model compute summary statistics , i.e quantiles and integrated quantiles.

2. For all pairs of models, compute statistics of absolute and relative tests by computing violation ratios.

3. Compute the variance of these statistics via bootstrapping.

4. Perform the hypothesis testing for all pairs models with a corrected confidence level taking into account the number of all pairs

5. Aggregate pairwise rankings to a rank using the Borda algorithm, that ranks the model by their number of wins in the stochastic dominance tests performed above.

## D  ABSOLUTE OR ALMOST STOCHASTIC DOMINANCE

**Almost FSD ($\varepsilon$-FSD)** Following Leshno & Levy (2002), Del Barrio et al. (2018) relaxed FSD via the violation ratio of FSD:

> **Definition 3** (FSD Violation Ratio (Del Barrio et al., 2018) ). *For $F_X \neq F_Y$ define the violation ratio:*
>
> $$\varepsilon_{\mathsf{W}_2}(F_X, F_Y) = \frac{\int_{A_0^{(1)}} (F_X^{(-1)}(t) - F_Y^{(-1)}(t))^2 dt}{\int_0^1 (F_X^{(-1)}(t) - F_Y^{(-1)}(t))^2 dt} = \frac{\int_0^1 (F_Y^{(-1)}(t) - F_X^{(-1)}(t))_+^2 \, dt}{\mathsf{W}_2^2(F_X, F_Y)},$$
>
> *where $A_0^{(1)} = \left\{ t \in (0,1) : F_Y^{(-1)}(t) > F_X^{(-1)}(t) \right\}$ is the violation set the relation $X \underset{FSD}{\succcurlyeq} Y$, and $\mathsf{W}_2$ is the Wasserstein$-2$ distance.*

Note that $0 \leq \varepsilon_{\mathsf{W}_2}(F_X, F_Y) \leq 1$, with value 0 if $X \underset{FSD}{\succ} Y$ and 1 if $Y \underset{FSD}{\succ} X$. For $\varepsilon \in (0, \frac{1}{2}]$, the relaxed FSD can be therefore defined as follows

$$X \underset{\varepsilon-\mathsf{FSD}}{\succcurlyeq} Y \iff \varepsilon_{\mathsf{W}_2}(F_X, F_Y) \leq \varepsilon. \tag{19}$$

Figure 5a in Appendix G illustrates $\varepsilon$-FSD, dashed areas represent the violation set.

**Almost SSD ($\varepsilon$-SSD)** Note that the original definition of $\varepsilon$-FSD of $X$ on $Y$ in Leshno & Levy (2002) is an $L_1$ definition and uses the CDF rather than quantiles: $\int_{-\infty}^{\infty} (F_X(x) - F_Y(x))_+ dx \leq \varepsilon \int_{\infty}^{\infty} |F_X(x) - F_Y(x)| dx$. Tzeng et al. (2013) gave a similar $L_1$ definition for $\varepsilon$-SSD using the second performance function $F^{(2)}(.)$. According to Tzeng et al. (2013), $X$ dominates $Y$ in the $\varepsilon$-SSD if $\int_{-\infty}^{\infty} (F_X^{(2)}(x) - F_Y^{(2)}(x))_+ dt \leq \varepsilon \int_{-\infty}^{+\infty} |F_X^{(2)}(x) - F_Y^{(2)}(x)| dx$. Following Del Barrio et al. (2018), we redefine $\varepsilon$-SSD using second quantiles and with a $L_2$ definition, this eases the analysis and practically the integration is on $(0, 1]$ rather than $(-\infty, \infty)$.

We define the SSD violation ratio as follows:

**Algorithm 1** Stochastic Order Multi-testing (relative and **absolute**)

1: **Input:** $F_1, ..., F_k$, $k$ models we want to rank corresponding to empirical measure $p_1 = \frac{1}{n} \sum_{i=1}^{n} \delta_{x_i^1}, \dots p_k = \frac{1}{n} \sum_{i=1}^{n} \delta_{x_i^k}$, **Threshold:** $\tau$.

2: **Input:** Desired stochastic order $\in \{1, 2\}$, $B$ number of bootstraps, $m = K^2$ number of comparisons, significance level $\alpha$.

3: **Cache the bootstraps samples and their statistics**

4: **for** $j = 1$ **to** $k$ **do**

5:     $p_j^0 \leftarrow p_j$

6:     **Get Quantiles and Integrated Quantiles**

7:     $Q_{0,j} \leftarrow \text{GETQUANTILES}(p_j)$

8:     $IQ_{0,j} \leftarrow \text{GETINTEGRATEDQUANTILES}(p_j)$

9:     **for** $b = 1$ **to** $B$ **do**

10:       **Get Quantiles and Integrated Quantiles**

11:       $p_j^b \leftarrow \text{RESAMPLEWITHREPLACEMENT}(p_j, n)$ {using quantiles and uniform}

12:       $Q_{b,j} \leftarrow \text{GETQUANTILES}(p_j^b)$

13:       $IQ_{b,j} \leftarrow \text{GETINTEGRATEDQUANTILES}(p_j^b)$

14:     **end for**

15: **end for**

16: **Compute all violation ratios**

17: $\varepsilon_{b,i,j} \leftarrow \text{COMPUTEVIOLATIONRATIOS}(F_i^b, F_j^b, \text{order})$ for $b = 0 \dots B$,   $i, j = 1 \dots k, i \neq j$ {ratio of Q or IQ of $j > i$ by total area}

18: $\varepsilon_{b,i,i} = 0, \forall b, i$

19: **Compute the sum statistics**

20: **for** $b = 0$ **to** $B$ **do**

21:     **for** $i = 1$ **to** $k$ **do**

22:       $\varepsilon_b^i \leftarrow \frac{1}{k-1} \sum_j \varepsilon_{b,i,j}$

23:     **end for**

24: **end for**

25: **Compute the relative statistics**

26: $\Delta \varepsilon_b^{i,j} = \varepsilon_b^i - \varepsilon_b^j, \forall b, i, j$

27: **Compute the Bootstrap Variance**

28: **for** $i = 1$ **to** $k$ **do**

29:     **for** $j = 1$ **to** $k$ **do**

30:       $\sigma_{ij} = \sqrt{\frac{1}{B-1} \sum_{b=1}^{B} (\Delta \varepsilon_b^{i,j} - \text{MEAN}(\Delta \varepsilon_b^{i,j}, b))^2}$

31:       $\sigma_{ij}^{\text{abs}} = \sqrt{\frac{1}{B-1} \sum_{b=1}^{B} (\varepsilon_{b,i,j} - \text{MEAN}(\varepsilon_{b,i,j}, b))^2}$

32:     **end for**

33: **end for**

34: **Compute the test**

35: $\text{Win}_{ij} = \text{Win}_{ij}^{\text{abs}} = 0$

36: **for** $i = 1$ **to** $k$ **do**

37:     **for** $j = 1$ **to** $k$ **do**

38:       **if** $i \neq j$ and $\Delta \varepsilon_0^{i,j} - \frac{1}{\sqrt{n}} \sigma_{ij} \Phi^{-1}(\alpha/k^2) \leq 0$ **then**

39:         $\text{Win}_{ij} = 1$ {with confidence level $1 - \alpha/k^2$}

40:       **end if**

41:       **if** $i \neq j$ and $\varepsilon_{0.i,j} - \frac{1}{\sqrt{n}} \sigma_{ij}^{\text{abs}} \Phi^{-1}(\alpha/k^2) \leq \tau$ **then**

42:         $\text{Win}_{ij}^{\text{abs}} = 1$ {with confidence level $1 - \alpha/k^2$}

43:       **end if**

44:     **end for**

45: **end for**

    rank = BORDA(Win) {with confidence level $1 - \alpha$}

    rank$_{\text{abs}}$ = BORDA(Win$^{\text{abs}}$) {with confidence level $1 - \alpha$}

46: **return** rank, rank$_{\text{abs}}$

---

**Algorithm 2** COMPUTEVIOLATIONRATIOS($F_a$,$F_b$,order)

> **if** order =1 **then**
>     **return** $\varepsilon_{W_2}(F_a, F_b)$ in Definition 3
> **else if** order=2 **then**
>     **return** $\varepsilon_{IQ}(F_a, F_b)$ in Definition 4
> **end if**

---

> **Definition 4** (SSD Violation Ratio ). *For $F_X \neq F_Y$ define the violation ratio:*
>
> $$\varepsilon_{IQ}(F_X, F_Y) = \frac{\int_{A_0^{(2)}}(F_X^{(-2)}(t) - F_Y^{(-2)}(t))^2 dt}{\int_0^1 (F_X^{(-2)}(t) - F_Y^{(-2)}(t))^2 dt} = \frac{\int_0^1 (F_Y^{(-2)}(t) - F_X^{(-2)}(t))_+^2 dt}{d_{IQ}^2(F_X, F_Y)},$$
>
> *where $A_0^{(2)} = \left\{ t \in (0,1) : F_Y^{(-2)}(t) > F_X^{(-2)}(t) \right\}$ is the violation set the relation $X \underset{SSD}{\succcurlyeq} Y$, and $d_{IQ}$ is the $L_2$ distance between the Integrated Quantiles $(F^{(-2)})$.*

We are now ready to define $\varepsilon$-SSD, for $\varepsilon \in (0, \frac{1}{2})$:

> $$X \underset{\varepsilon - \text{SSD}}{\succcurlyeq} Y \iff \varepsilon_{IQ}(F_X, F_Y) \leq \varepsilon \tag{20}$$

Figure 5b in Appendix G illustrates the second order, dashed areas represent the violation set of SSD of $X$ on $Y$. Integrated quantiles fully characterize one dimensional distributions as can be seen from the Theorem 3 stated and proved in Appendix I:

## E   RELATED WORKS ON STOCHASTIC DOMINANCE

**Stochastic Dominance** In Dror et al. (2018; 2019); Ulmer et al. (2022); Simpson (2021) a distributional assessment of the models based on stochastic dominance was proposed to overcome the limitations of the ubiquitous Mean-Variance Risk model used in machine learning.

(Ulmer et al., 2022) used first order almost stochastic dominance and advocated for selecting a model $A$ over $B$ based on a metric $m_i$ if: $M_i|A, X \underset{\varepsilon - \text{FSD}}{\succcurlyeq} M_i|B, X$. We expand this to the Relative-FSD. In natural language (and other) applications, it is often crucial to mitigate the risk of outputs with low metrics, especially when those metrics quantify important socio-technical guardrails such as model's toxicity, safety, or robustness. Unfortunately, the first stochastic ordering does not capture an assessment of the left tail behavior of $M_i|A, X$ and $M_i|B, X$ and hence does not provide a risk-aware assessment Ogryczak & Ruszczynski (2002). To alleviate this issue, we instead consider the *second* order stochastic ordering and use our second order *almost* or *relative* stochastic dominance tests introduced in Section 3 for selecting a model A if: $M_i|A, X \underset{\varepsilon \text{ or } R-\text{SSD}}{\succcurlyeq} M_i|B, X$.

## F   SUPPLEMENT DISCUSSIONS

### F.1   MEAN RISK MODELS

| Name | Risk Measure | $\alpha-$ consistency with SSD |
|---|---|---|
| Standard deviation | $\sigma_X = \sqrt{\mathbb{E}(X - \mu_X)^2}$ | not consistent |
| Absolute semi deviation | $\delta_X = \mathbb{E}(\mu_X - X)_+$ | $1-$ consistent |
| Negative Tail Value at Risk | $-\text{TVAR}_X(p) = -\frac{F^{(-2)}(p)}{p}$ | $1-$ consistent for all $p \in (0,1]$ |
| Mean absolute deviation from a quantile | $h_X(p) = \mu_x - \frac{F_X^{(-2)}(p)}{p}$ | $1-$ consistent for all $p \in (0,1]$ |
| Gini Tail | $\Gamma_X = 2\int_0^1 (\mu_X p - F_X^{(-2)}(p))dp$ | $1-$ consistent |

Table 1: Risk models and their $\alpha-$consistency with SSD.

Note that several risks in Table 1 use the second quantile function as part of an assessment of the left tails of the outcomes.

## F.2   $\delta-$ Consistency of Gini-Risk Models with $\varepsilon$-SSD

$\delta-$ **Consistency of Gini-Risk Models with $\varepsilon$-SSD**   We relax the definition of $\alpha-$ consistency of mean-risk models with SSD to $(\alpha, \delta)$ consistency with $\varepsilon$-SSD as follows:

**Definition 5** (($\alpha, \delta$) consistency of MRM with $\varepsilon$-SSD)**.** *A mean-risk model $(\mu_X, r_X)$ is $(\alpha, \delta)$ consistent with $\varepsilon$-SSD, if there exists $\alpha, \delta > 0$ such that $X \underset{\varepsilon\text{-}SSD}{\succcurlyeq} Y \implies \mu_X - \alpha r_x + \delta \geq \mu_Y - \alpha r_Y$*

It is easy to see that the Mean-Gini tail MRM of $X$ and $Y$ is consistent with their $\varepsilon$-SSD:

**Proposition 1.** *The Mean-Gini Tail MRM is $(1, 2\varepsilon^{\frac{1}{2}} d_{IQ}(F_X, F_Y))$ consistent with $\varepsilon$-SSD.*

*Proof of Proposition 1.*

$$\mu_X - \Gamma_X = \mu_X - 2\int_0^1 (\mu_X p - F_X^{(-2)}(p))dp = 2\int_0^1 (F_X^{(-2)}(p) - F_Y^{(-2)}(p) + F_Y^{(-2)}(p))dp$$

$$= 2\int_0^1 F_Y^{(-2)}(p) + 2\int_{A_0^{(2)}} (F_X^{(-2)}(p) - F_Y^{(-2)}(p))dp + 2\underbrace{\int_{[0,1]/A_0^{(2)}} (F_X^{(-2)}(p) - F_Y^{(-2)}(p))dp}_{\geq 0}$$

$$\geq 2\int_0^1 F_Y^{(-2)}(p) - 2\int_{A_0^{(2)}} |F_X^{(-2)}(p) - F_Y^{(-2)}(p))|dp$$

$$= \mu_Y - \Gamma_Y - 2\int_0^1 (F_Y^{(-2)}(p) - F_X^{(-2)}(p))_+ dp$$

$$\geq \mu_Y - \Gamma_Y - 2\left(\int_0^1 dp\right)^{\frac{1}{2}} \left(\int_0^1 (F_Y^{(-2)}(p) - F_X^{(-2)}(p))_+^2 dp\right)^{\frac{1}{2}} \text{(Cauchy-Schwartz)}$$

$$\geq \mu_Y - \Gamma_Y - 2\varepsilon^{\frac{1}{2}} d_{IQ}(F_X, F_Y) \text{(By assumption } X \underset{\varepsilon-\text{SSD}}{\succcurlyeq} Y)$$

$\square$

## F.3   Rank Aggregation

Given $N$ ranks $\pi_i, i = 1 \ldots N$ represented as permutations in $S_k$, the rank aggregation in (Pihur et al., 2009) solves the following problem :

$$\min_{\pi \in S_k} \sum_{i=1}^N \alpha_i d(\pi, \pi_i),$$

where $\alpha_i \geq 0, \sum_{i=1}^N \alpha_i = 1$ represent importance of each ranking and $d$ is a distance between permutations. (Pihur et al., 2009) have multiple choices of distance such as Pearson or Kendall's-Tau. We fixed through out our experiments the distance to Pearson.

## F.4   Mean Win Rate and CDF normalizers in portfolio

To unpack the notations in (17), consider a distribution $\mathcal{A}$ on models space. For a sample $X \sim D_i$ and a model $A \sim \mathcal{A}$, the metric $m_i()$ normalization through its CDF can be written as follows:

$$F_{M_i}(m_i(A(X)) = \mathbb{E}_{B \sim \mathcal{A}} \mathbb{E}_{Y \sim D_i} \mathbb{1}_{m_i(B(Y)) \leq m_i(A(X))}. \tag{21}$$

Hence for a model $A$ on each evaluated sample the CDF normalizer computes a soft ranking of the evaluation of the model $A$ with a metric $m_i$ on the sample $X$ with respect to all models and all samples.

**Remark 1** (Mean Win Rate ). *Note that in LLM leaderborads such as HELM and Hugging face, the performance of a model $A$ evaluated with a metric $m_i$, is summarized via a Mean Win Rate (MWR) aggregated on models level looking on expected metrics*

$$\text{MWR}_{A,M_i} = \mathbb{E}_{B\sim\mathcal{A}}\mathbb{1}_{\mathbb{E}_{X\sim D_i}[m_i(B(X))]\leq\mathbb{E}_{X\sim D_i}[m_i(A(X))]}, \tag{22}$$

*or aggregated on sample level marginalizing on models with a* max*:*

$$\overline{\text{MWR}}_{A,M_i} = \mathbb{E}_{X\sim D_i}\mathbb{1}_{\max_{B\neq A} m_i(B(X))\leq m_i(A(X))}, \tag{23}$$

*Contrasting* (21) *,* (22) *and* (23) *we see that instead of looking at the MWR summary statistics that does not allow to consider all order statistics and relative ordering as well the risks of tails events, we consider a full distributional assessment in the metrics portfolio approach.*

## G FIGURES

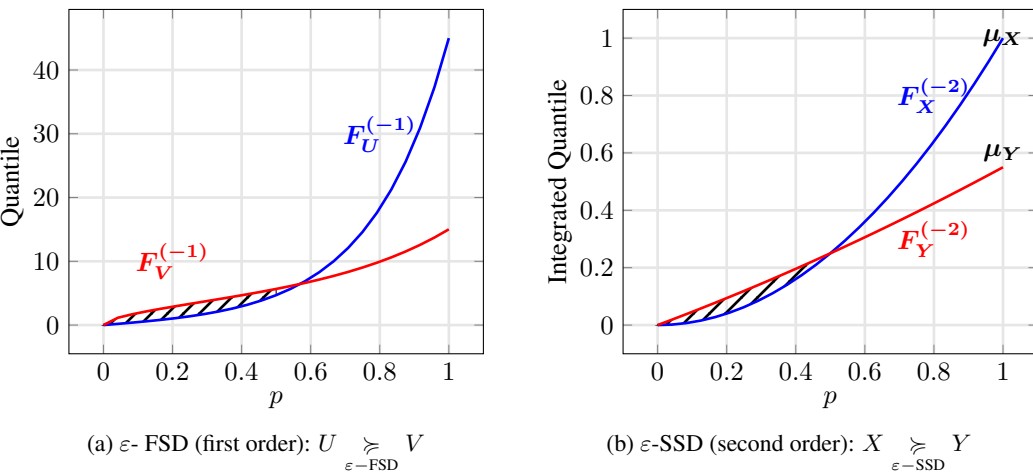

(a) $\varepsilon$- FSD (first order): $U \underset{\varepsilon-\text{FSD}}{\succcurlyeq} V$      (b) $\varepsilon$-SSD (second order): $X \underset{\varepsilon-\text{SSD}}{\succcurlyeq} Y$

Figure 5: **(a) An Example of Almost First Order Stochastic Dominance:** Plots of quantile functions of $U$ and $V$. Dashed areas is the violation set of first order stochastic dominance of $U$ on $V$. **(b) An Example of Almost Second Order Stochastic Dominance:** Plots of integrated quantile functions; dashed area is the violation set for the second order stochastic dominance of $X$ on $Y$.

## H CENTRAL LIMIT THEOREMS

### H.1 CLT FOR $\varepsilon$-SSD

> **Theorem 1** (Central Limit Theorem for $\varepsilon$-SSD). *Assume that $F_X$, $F_Y$ are supported on intervals[a] in $[-M, M]$, and have pdfs $f_x, f_y$ such that $\frac{f_x'(p)}{f_x^3(p)}$, $\frac{f_y'(p)}{f_y^3(p)}$ are bounded almost everywhere on the support of $f_x$ and $f_y$ respectively. Assume we have $n$ samples from $F_X$ and $m$ samples from $F_Y$, with $n, m \to \infty$ such that $\frac{n}{n+m} \to \lambda$ for some $\lambda$. Then*
>
> $$\sqrt{\frac{mn}{m+n}} \left(\varepsilon_{IQ}(F_X^n, F_Y^m) - \varepsilon_{IQ}(F_X, F_Y)\right) \to \mathcal{N}(0, \sigma_\lambda^2(F_X, F_Y))$$
>
> *where*
>
> $$\sigma_\lambda^2(F_X, F_Y) = \frac{1}{d_{IQ}^8(F_X, F_Y)} \left[(1-\lambda)\text{Var}(v_X(U)) + \lambda\text{Var}(v_Y(U))\right],$$
>
> *for $U \sim \text{Unif}[0, 1]$, $v_Y(t) = 2\left(\frac{1}{f_y(F_Y^{-1}(t))}\right)\left(\int_t^1 (F_X^{(-2)}(p) - F_Y^{(-2)}(p))_+ dp\right)$, and $v_X(t) = 2\left(\frac{1}{f_x(F_X^{-1}(t))}\right)\left(\int_t^1 (F_X^{(-2)}(p) - F_Y^{(-2)}(p))_- dp\right)$.*
>
> ---
> [a]The interval for $F_X$ and for $F_Y$ need not coincide.

**Remark 2** (Non-independent samples). *Theorem 1 assumes that the $n$-sample from $F_X$ is independent of the $m$-sample for $F_Y$. Consider instead the setting where there are $n$ samples from $F_X$ and $F_Y$ that are dependent (e.g. $X, Y$ are evaluations of different models applied to the same data). We can describe general dependence structure as the following. Suppose $(X, Y)$ has marginals $X \sim F_X, Y \sim F_Y$, with some unknown dependence structure (optionally described by the copula $C_{XY}(u_x, u_y) = \Pr(F_X(X) \le u_x, F_Y(Y) \le u_y)$). Let*

$$(U_x, U_y) = (F_X(X), F_Y(Y)) \sim C_{XY}.$$

*Note that $U_x$ and $U_y$ have marginals equal to $\mathrm{Unif}([0, 1])$, but $U_x$ and $U_y$ may be dependent. Hence the variances in each term of the decomposition (25) in the appendix cannot be added. Instead, one should modify the result of Theorem 1 to use*

$$\bar{\sigma}_\lambda^2(F_X, F_Y) = \frac{1}{d_{IQ}^8(F_X, F_Y)} \mathrm{Var}\left[v_X(U_x) + v_Y(U_y)\right].$$

## H.2 CLT FOR RELATIVE STATISTICS

We focus here on presenting the Central Limit Theorem for SSD. The relative FSD has a similar form and we omit its statement here.

---

**Theorem 2** (Central limit Theorem for Relative SSD). *Assume $F_{1n}, \ldots, F_{kn}$ are available and independent, and each $F_i$ satisfies the conditions of Theorem 1. Then*

$$\sqrt{n}\left(\Delta\varepsilon_{i_1,i_2}^{(2)}(F_n) - \Delta\varepsilon_{i_1,i_2}^{(2)}(F)\right) \to_w \mathcal{N}\left(0, \frac{1}{(k-1)^2}\sum_{i=1}^k \sigma_i^2(i_1, i_2)\right).$$

*where*

$$\sigma_i^2(i_1, i_2) = \begin{cases} \mathrm{Var}\left(\frac{2v_{i_1 i_2}^{(1)-}(U_i)}{d_{IQ}^4(F_{i_1}, F_{i_2})} + \sum_{j \ne i_1, i_2} \frac{v_{i_1 j}^{(1)-}(U_i)}{d_{IQ}^4(F_{i_1}, F_j)}\right) & i = i_1 \\ \mathrm{Var}\left(\frac{2v_{i_1 i_2}^{(2)+}(U_i)}{d_{IQ}^4(F_{i_1}, F_{i_2})} - \sum_{j \ne i_1, i_2} \frac{v_{i_2 j}^{(1)-}(U_i)}{d_{IQ}^4(F_{i_2}, F_j)}\right) & i = i_2 \\ \mathrm{Var}\left(\frac{v_{i_1 j}^{(2)+}(U_i)}{d_{IQ}^4(F_{i_1}, F_j)} - \frac{v_{i_2 j}^{(2)+}(U_i)}{d_{IQ}^4(F_{i_2}, F_j)}\right) & i \ne i_1, i_2 \end{cases}$$

*for $U_i \sim \mathrm{Unif}([0, 1])$ all independent, and $v_{ij}^{(1),-}(t) = 2\left(\frac{dF_i^{-1}(t)}{dt}\right)\left(\int_t^1 (F_i^{(-2)}(p) - F_j^{(-2)}(p))_- dp\right), v_{ij}^{(2),+}(t) = 2\left(\frac{dF_j^{-1}(t)}{dt}\right)\left(\int_t^1 (F_i^{(-2)}(p) - F_j^{(-2)}(p))_+ dp\right).$*

---

**Remark 3** (Dependent samples). *If the $F_{in}$ are dependent, a similar expression to that shown in Remark 2 for the absolute testing case also holds here. The statement is omitted.*

## I PROOF OF THEOREM 3

---

**Theorem 3** ($d_{IQ}$ is a metric). *$d_{IQ}$ is a metric on the space of univariate distributions with continuous CDF, moreover, it metrizes the weak topology.*

---

First, we show that $d_{IQ}(F, G) = 0$ if and only if $F = G$. The forward direction is obvious. For the reverse direction, if $d_{IQ}(F, G) = 0$, then $F^{(-2)}(t) = G^{(-2)}(t)$ a.e. By the continuity of integrated quantiles, this implies $F^{(-2)} = G^{(-2)}$ everywhere. Then, since $F^{(-1)}(t)$ is simply the derivative of $F^{(-2)}(t)$ with respect to $t^{[6]}$, $F^{(-1)} = G^{(-1)}$ everywhere by differentiating both sides of $F^{(-2)}(t) = G^{(-2)}(t)$. Hence $F = G$ since distributions are uniquely determined by their quantile functions.

---

[6]This follows because $F^{-2}$ is the integral of the finite-valued quantile function $F^{-1}(t)$.

The triangle inequality follows from the triangle inequality of the $L_2$ norm, since $\sqrt{\int_0^1 (F^{(-2)}(t) - G^{(-2)}(t))^2 dt} = \|F^{(-2)}(t) - G^{(-2)}(t)\|_{L_2([0,1])}$. Hence $d_{IQ}$ is a metric. By Theorem 10 in Gushchin & Borzykh (2017), we know that random variable $X_{(i)} \to_w X$ (with cdf $F_{(i)}$) if and only if $F_{(i)}^{(-2)}$ converges uniformly to $F^{(-2)}$. Hence $d_{IQ}$ must metrize weak convergence.

## J PROOFS OF CENTRAL LIMIT THEOREMS

### J.1 ABSOLUTE TESTING: PROOF OF THEOREM 1

Note that for $U_i$ and $V_i$ an $n$-sample and an $m$-sample respectively from $\mathrm{Unif}([0,1])$, we can get $X_i, Y_i$ as $X_i = F^{-1}(U_i)$, $Y_i = G^{-1}(V_i)$. Let $H_{n,1}$ and $H_{m,2}$ be the empirical d.f.s of the $U_i$ and $V_i$ respectively. We have

$$F_n^{-1}(t) = F^{-1}(H_{n,1}^{-1}(t)),$$

hence

$$F_n^{(-2)}(t) = \int_0^t F_n^{-1}(p)dp = \int_0^t F^{-1}(H_{n,1}^{-1}(p))dp.$$

We are interested in

$$\varepsilon_{IQ}(F_n, G_m) = \frac{\int_{A_0}(F_n^{(-2)}(t) - G_m^{(-2)}(t))^2 dt}{d_{IQ}^2(F_n, G_m)},$$

where

$$A_0 = \left\{ t \in (0,1) : G_m^{(-2)}(t) > F_n^{(-2)}(t) \right\},$$

is the violation set.

It is shown in Gushchin & Borzykh (2017) (Theorem 10 therein) that integrated quantiles converge uniformly, i.e. $F_n^{(-2)}(t) \to F^{(-2)}(t)$ pointwise. As an immediate consequence, we have

$$\varepsilon_{IQ}(F_n, G_m) \to_{a.s.} \varepsilon_{IQ}(F, G).$$

We apply the following decomposition and bound the two terms separately:

$$\varepsilon_{IQ}(F_n, G_m) - \varepsilon_{IQ}(F, G) = (\varepsilon_{IQ}(F_n, G_m) - \varepsilon_{IQ}(F, G_m)) + (\varepsilon_{IQ}(F, G_m) - \varepsilon_{IQ}(F, G)). \quad (24)$$

We derive asymptotic normality of these terms for $G_m$, the proof for $F_n$ is identical by symmetry.

We introduce the statistics

$$S_m = \int_0^1 (F^{(-2)}(t) - G_m^{(-2)}(t))^2 dt$$

$$S_m^+ = \int_0^1 (F^{(-2)}(t) - G_m^{(-2)}(t))_+^2 dt$$

$$S_m^- = \int_0^1 (F^{(-2)}(t) - G_m^{(-2)}(t))_-^2 dt$$

The nonrandom $S, S^+, S^-$ are defined similarly with $G$ instead of $G_m$.

Next, set

$$T_m = \sqrt{m}(S_m - S)$$
$$T_m^+ = \sqrt{m}(S_m^+ - S^+)$$
$$T_m^- = \sqrt{m}(S_m^- - S^-).$$

**Theorem 4.** *Assume that $G$ is supported on an interval that is a subset of $[-M, M]$, and has pdf $g$ such that $\frac{g'(p)}{g^3(p)}$ is bounded almost everywhere on the support of $g$. Then*

$$T_m = \alpha_{m,2}(v) + o_P(1)$$

$$T_m^+ = \alpha_{m,2}(v^+) + o_P(1)$$

$$T_m^- = \alpha_{m,2}(v^-) + o_P(1)$$

where we define $\alpha_{m,2}(t) = \sqrt{m}(t - H_{m,1}^{-1}(t))$ and $\alpha_{m,2}(v) = \int_0^1 v(t)\alpha_{m,2}(t)dt$, and

$$v(t) = 2\left(\frac{1}{g(G^{-1}(t))}\right)\left(\int_t^1 F^{(-2)}(p) - G^{(-2)}(p)dp\right).$$

$$v^+(t) = 2\left(\frac{1}{g(G^{-1}(t))}\right)\left(\int_t^1 (F^{(-2)}(p) - G^{(-2)}(p))_+ dp\right),$$

$$v^-(t) = 2\left(\frac{1}{g(G^{-1}(t))}\right)\left(\int_t^1 (F^{(-2)}(p) - G^{(-2)}(p))_- dp\right).$$

*Proof.* We begin with $T_m$. Note that[7]

$$T_m = \sqrt{m}(S_m - S)$$

$$= \sqrt{m}\int_0^1 (F^{(-2)}(t) - G_m^{(-2)}(t))^2 - (F^{(-2)}(t) - G^{(-2)}(t))^2 dt$$

$$= \sqrt{m}\int_0^1 \left[2F^{(-2)}(t) - G_m^{(-2)}(t) - G^{(-2)}(t)\right](G^{(-2)}(t) - G_m^{(-2)}(t))dt$$

$$\to 2\sqrt{m}\int_0^1 \left[F^{(-2)}(t) - G^{(-2)}(t)\right](G^{(-2)}(t) - G_m^{(-2)}(t))dt$$

$$= 2\sqrt{m}\int_0^1 \left[F^{(-2)}(t) - G^{(-2)}(t)\right]\left[\int_0^t G^{(-1)}(p) - G^{(-1)}(H_{m,1}^{-1}(p)))dp\right]dt$$

Let us do integration by parts:

$$2\sqrt{m}\int_0^1 \left[F^{(-2)}(t) - G^{(-2)}(t)\right]\left[\int_0^t G^{(-1)}(p) - G^{(-1)}(H_{m,1}^{-1}(p)))dp\right]dt =$$

$$= 2\sqrt{m}\left[\left(\int_0^1 F^{(-2)}(t) - G^{(-2)}(t)dt\right)\left[\int_0^1 G^{(-1)}(t) - G^{(-1)}(H_{m,1}^{-1}(t)))dt\right]\right.$$

$$\left. - \int_0^1 \left(\int_0^t F^{(-2)}(p) - G^{(-2)}(p)dp\right)\left[G^{(-1)}(t) - G^{(-1)}(H_{m,1}^{-1}(t)))\right]dt\right]$$

$$= 2\sqrt{m}\int_0^1 \left(\int_t^1 F^{(-2)}(p) - G^{(-2)}(p)dp\right)\left[G^{(-1)}(t) - G^{(-1)}(H_{m,1}^{-1}(t)))\right]dt$$

$$= 2\sqrt{m}\int_0^1 \left(\frac{dG^{-1}(t)}{dt}\right)\left(\int_t^1 F^{(-2)}(p) - G^{(-2)}(p)dp\right)(t - H_{m,1}^{-1}(t))dt$$

$$+ O\left(\sqrt{m}\int_0^1 \int_t^1 F^{(-2)}(p) - G^{(-2)}(p)dp)(t - H_{m,1}^{-1}(t))^2 dt\right)$$

$$= 2\sqrt{m}\int_0^1 \left(\frac{dG^{-1}(t)}{dt}\right)\left(\int_t^1 F^{(-2)}(p) - G^{(-2)}(p)dp\right)(t - H_{m,1}^{-1}(t))dt + o_P(1).$$

In the penultimate step we have used a first-order Taylor series on $G^{-1}(t)$ via the assumption that $\frac{d^2 G^{-1}(t)}{dt^2} = -\frac{g'(G^{-1}(t))}{g^3(G^{-1}(t))}$ is bounded almost everywhere, and in the final step we have noted that

$$\sqrt{m}\int_0^1 \left(\int_t^1 F^{(-2)}(p) - G^{(-2)}(p)dp\right)(t - H_{m,1}^{-1}(t))^2 dt \le 2\sqrt{m}\int_0^1 (t - H_{m,1}^{-1}(t))^2 dt$$

$$= o_P(1),$$

since the support of $F$ and $G$ lie in $[-M, M]$ and $\int_0^1 (t - H_{m,1}^{-1}(t))^2 dt = O_p(1/m)$.

---

[7]Convergence here is uniform convergence of the integrated quantiles.

We then have
$$T_m = \alpha_{m,2}(v) + o_P(1),$$
where $\alpha_{m,2}(t) = \sqrt{m}(t - H_{m,1}^{-1}(t))$, and $\alpha_{m,2}(v) = \int_0^1 v(t)\alpha_{m,2}(t)dt$ where

$$v(t) = 2\left(\frac{dG^{-1}(t)}{dt}\right)\left(\int_t^1 F^{(-2)}(p) - G^{(-2)}(p)dp\right).$$

Similarly,
$$T_m^+ = \alpha_{m,2}(v^+) + o_P(1), \quad T_m^- = \alpha_{m,2}(v^-) + o_P(1)$$
where

$$v^+(t) = 2\left(\frac{dG^{-1}(t)}{dt}\right)\left(\int_t^1 (F^{(-2)}(p) - G^{(-2)}(p))_+dp\right),$$

$$v^-(t) = 2\left(\frac{dG^{-1}(t)}{dt}\right)\left(\int_t^1 (F^{(-2)}(p) - G^{(-2)}(p))_-dp\right).$$

$\square$

**Corollary 1.** *Assume that $G$ is supported on an interval in $[-M, M]$, and has pdf $g$ such that $\frac{g'(p)}{g^3(p)}$ is bounded almost everywhere on the support of $g$. Then as $m \to \infty$*

$$\sqrt{m}(\epsilon_{IQ}(F, G_m) - \epsilon_{IQ}(F, G)) \to_w \mathcal{N}(0, \sigma^2)$$

*and if additionally $n \to \infty$*

$$\sqrt{m}(\epsilon_{IQ}(F_n, G_m) - \epsilon_{IQ}(F_n, G)) \to_w \mathcal{N}(0, \sigma^2),$$

*if for $U \sim \text{Unif}([0, 1])$*

$$\sigma^2 = \frac{\text{Var}(v^+(U))}{d_{IQ}^8(F, G)}$$

*is finite.*

*Proof.* Note that by Theorem 4

$$\sqrt{m}(\epsilon_{IQ}(F, G_m) - \epsilon_{IQ}(F, G)) = \sqrt{m}\left(\frac{S_m^-}{S_m} - \frac{S^-}{S}\right) = \frac{\sqrt{m}}{SS_m}(T_m^- - T_m) \to -\frac{\alpha_{m,2}(v^+)}{S^2}$$

since $S_m \to S$ a.s. Recalling the definition of $\alpha_{m,2}$ yields asymptotic normality with zero mean as in Del Barrio et al. (2018), and variance as calculated in the corollary statement.

The case of $\sqrt{m}(\epsilon_{IQ}(F_n, G_m) - \epsilon_{IQ}(F_n, G))$ follows similarly since integrated quantiles weakly converge as $F_n \to F$. $\square$

Continuing with the main proof, recalling (24) and using Corollary 1 along with the asymptotic independence of the two terms and the fact that $\frac{n}{n+m} \to \lambda$, we have

$$\sqrt{\frac{mn}{m+n}}\left(\varepsilon_{IQ}(F_n, G_m) - \varepsilon_{IQ}(F, G)\right)$$
$$= \sqrt{(1-\lambda)n}(\varepsilon_{IQ}(F_n, G_m) - \varepsilon_{IQ}(F, G_m)) + \sqrt{\lambda n}(\varepsilon_{IQ}(F, G_m) - \varepsilon_{IQ}(F, G)) \quad (25)$$
$$\to \mathcal{N}(0, \sigma_\lambda^2(F, G))$$

where

$$\sigma_\lambda^2(F, G) = \frac{1}{d_{IQ}^8(F, G)}\left[(1-\lambda)\text{Var}(v_F(U)) + \lambda\text{Var}(v_G(U))\right].$$

Here, we have defined

$$v_G(t) = 2\left(\frac{1}{g(G^{-1}(t))}\right)\left(\int_t^1 (F^{(-2)}(p) - G^{(-2)}(p))_+dp\right),$$

and

$$v_F(t) = 2\left(\frac{1}{f(F^{-1}(t))}\right)\left(\int_t^1 (F^{(-2)}(p) - G^{(-2)}(p))_-dp\right).$$

## J.2 RELATIVE TESTING: PROOF OF THEOREM 2

Note that

$$
\begin{aligned}
\Delta\varepsilon_{IQ}^{i_1,i_2}(F) &= \epsilon_{IQ}^{i_1}(F) - \epsilon_{IQ}^{i_2}(F) \\
&= \frac{1}{k-1}\left[\sum_{j\neq i_1}\epsilon_{IQ}^{i_1 j} - \sum_{j\neq i_2}\epsilon_{IQ}^{i_2 j}\right] \\
&= \frac{1}{k-1}\left[2\epsilon_{IQ}^{i_1 i_2} - 1 + \sum_{j\neq i_1,i_2}(\epsilon_{IQ}^{i_1 j} - \epsilon_{IQ}^{i_2 j})\right].
\end{aligned}
$$

For compactness, let us introduce the differencing notation $\phi(\cdot)|_F^{F_n} = \phi(F_n) - \phi(F)$. We seek a CLT on

$$
\begin{aligned}
\sqrt{n}(\widehat{\Delta\varepsilon_{IQ}^{i_1,i_2}}(F_n) - \Delta\varepsilon_{IQ}^{i_1,i_2}(F)) &= \frac{\sqrt{n}}{k-1}\left(2\epsilon_{IQ}(\cdot,F_{i_2,n}) + \sum_{j\neq i_1,i_2}\epsilon_{IQ}(\cdot,F_{j,n})\right)\Bigg|_{F_{i_1}}^{F_{i_1,n}} \\
&\quad + \frac{\sqrt{n}}{k-1}\left(2\epsilon_{IQ}(F_{i_1},\cdot) - \sum_{j\neq i_1,i_2}\epsilon_{IQ}(\cdot,F_{j,n})\right)\Bigg|_{F_{i_2}}^{F_{i_2,n}} \\
&\quad + \frac{\sqrt{n}}{k-1}\sum_{j\neq i_1,i_2}(\epsilon_{IQ}(F_{i_1},\cdot) - \epsilon_{IQ}(F_{i_2},\cdot))|_{F_j}^{F_{j,n}} \\
\to_w \underbrace{\frac{\sqrt{n}}{k-1}\left(2\epsilon_{IQ}(\cdot,F_{i_2}) + \sum_{j\neq i_1,i_2}\epsilon_{IQ}(\cdot,F_j)\right)\Bigg|_{F_{i_1}}^{F_{i_1,n}}}_{I} &+ \underbrace{\frac{\sqrt{n}}{k-1}\left(2\epsilon_{IQ}(F_{i_1},\cdot) - \sum_{j\neq i_1,i_2}\epsilon_{IQ}(\cdot,F_j)\right)\Bigg|_{F_{i_2}}^{F_{i_2,n}}}_{II} \\
&\quad + \underbrace{\frac{\sqrt{n}}{k-1}\sum_{j\neq i_1,i_2}(\epsilon_{IQ}(F_{i_1},\cdot) - \epsilon_{IQ}(F_{i_2},\cdot))|_{F_j}^{F_{j,n}}}_{III}
\end{aligned}
$$

where we have used the uniform convergence of integrated quantiles. Note that $I$, $II$, and each term in the sum in $III$ are all independent.

Define

$$
v_{ij}^{(1)}(t) = 2\left(\frac{dF_i^{-1}(t)}{dt}\right)\left(\int_t^1 F_i^{(-2)}(p) - F_j^{(-2)}(p)dp\right),
$$

$$
v_{ij}^{(2)}(t) = 2\left(\frac{dF_j^{-1}(t)}{dt}\right)\left(\int_t^1 F_i^{(-2)}(p) - F_j^{(-2)}(p)dp\right),
$$

and $v_{ij}^{(1)+}$, $v_{ij}^{(2)+}$ similarly. Then by the proof of Corollary 1, each term in $III$ converges to

$$
\begin{aligned}
\frac{\sqrt{n}}{k-1}(\epsilon_{IQ}(F_{i_1},\cdot) - \epsilon_{IQ}(F_{i_2},\cdot))|_{F_j}^{F_{j,n}} &\to -\frac{\alpha_{m,j}(v_{i_1 j}^{(2)+})}{(k-1)d_{IQ}^4(F_{i_1},F_j)} + \frac{\alpha_{m,j}(v_{i_2 j}^{(2)+})}{(k-1)d_{IQ}^4(F_{i_2},F_j)} \\
&= \frac{1}{k-1}\alpha_{m,j}\left(-\frac{v_{i_1 j}^{(2)+}}{d_{IQ}^4(F_{i_1},F_j)} + \frac{v_{i_2 j}^{(2)+}}{d_{IQ}^4(F_{i_2},F_j)}\right) \\
&\to_w \mathcal{N}\left(0, \frac{1}{(k-1)^2}\sigma_j^2(i_1,i_2)\right), \quad \forall j\neq i_1,i_2.
\end{aligned}
$$

where

$$
\sigma_j^2(i_1,i_2) = \frac{1}{(k-1)^2}\mathrm{Var}\left(\frac{v_{i_1 j}^{(2)+}(U)}{d_{IQ}^4(F_{i_1},F_j)} - \frac{v_{i_2 j}^{(2)+}(U)}{d_{IQ}^4(F_{i_2},F_j)}\right), \quad \forall j\neq i_1,i_2,
$$

and $U \sim \text{Unif}([0,1])$. Similarly for $I$ and $II$,

$$I \to_w \mathcal{N}\left(0, \frac{1}{(k-1)^2}\sigma_{i_1}^2(i_1, i_2)\right)$$

$$II \to_w \mathcal{N}\left(0, \frac{1}{(k-1)^2}\sigma_{i_2}^2(i_1, i_2)\right)$$

where[8]

$$\sigma_{i_1}^2(i_1, i_2) = \text{Var}\left(\frac{2v_{i_1 i_2}^{(1)-}(U)}{d_{IQ}^4(F_{i_1}, F_{i_2})} + \sum_{j \neq i_1, i_2}\frac{v_{i_1 j}^{(1)-}(U)}{d_{IQ}^4(F_{i_1}, F_j)}\right),$$

$$\sigma_{i_2}^2(i_1, i_2) = \text{Var}\left(\frac{2v_{i_1 i_2}^{(2)+}(U)}{d_{IQ}^4(F_{i_1}, F_{i_2})} - \sum_{j \neq i_1, i_2}\frac{v_{i_2 j}^{(1)-}(U)}{d_{IQ}^4(F_{i_2}, F_j)}\right).$$

Putting everything together via independence,

$$\sqrt{n}\left(\widehat{\Delta\varepsilon_{IQ}^{i_1, i_2}}(F_n) - \Delta\varepsilon_{IQ}^{i_1, i_2}(F)\right) \to_w \mathcal{N}\left(0, \frac{1}{(k-1)^2}\sum_{i=1}^{k}\sigma_i^2(i_1, i_2)\right).$$

## K  CONSISTENCY OF BOOTSTRAPPING

In this section, we consider the relaxation measure using the CDFs[9]:

$$\tilde{\epsilon}_\ell(F_X, F_Y) = \frac{\int_{-\infty}^{\infty}(F_Y^{(\ell)}(t) - F_X^{(\ell)}(t))_+^2\,dt}{\int_{-\infty}^{\infty}(F_Y^{(\ell)}(t) - F_X^{(\ell)}(t))^2\,dt}.$$

Note that we can relax FSD as follows:

$$Y \underset{\varepsilon-\text{FSD}}{\succeq} X \iff \tilde{\epsilon}_1(F_X, F_Y) \leq \varepsilon. \tag{26}$$

Similarly we can relax SSD as follows:

$$Y \underset{\varepsilon-\text{SSD}}{\succeq} X \iff \tilde{\epsilon}_2(F_X, F_Y) \leq \varepsilon. \tag{27}$$

We will prove bootstrap consistency for $\ell = 1$ (approximate first order dominance), the proof for $\ell = 2$ (approximate second order dominance) is similar.

We seek to show that the bootstrapped variance $\text{Var}(\tilde{\epsilon}_1(F_X^{n*}, F_Y^{m*}))$ is an asymptotically consistent estimator of $\text{Var}(\tilde{\epsilon}_1(F_X^n, F_Y^m))$, i.e. their ratio goes to 1:

$$\frac{\text{Var}(\tilde{\epsilon}_1(F_X^{n*}, F_Y^{m*}))}{\text{Var}(\tilde{\epsilon}_1(F_X^n, F_Y^m))} \to_p 1.$$

Note we can write this as

$$\frac{\text{Var}(\tilde{\epsilon}_1(F_X^{n*}, F_Y^{m*})}{\text{Var}(\tilde{\epsilon}_1(F_X^n, F_Y^m))} \to_p \frac{\text{Var}(T(F_X^{n*}, F_Y^{m*}))}{\text{Var}(T(F_X^n, F_Y^m))},$$

where

$$T(F_X, F_Y) = \frac{\int_{-\infty}^{\infty}(F_Y(t) - F_X(t))_+^2\,dt}{\int_{-\infty}^{\infty}(F_Y(t) - F_X(t))^2\,dt}.$$

---

[8]This $U \sim \text{Unif}([0,1])$ is drawn simply for this variance calculation and is not dependent on anything outside of this equation.

[9]The result using quantiles as described in the main text is less straightforward and if left for future work.

Consider the metric created by the sup norm

$$\rho_\infty(F, G) = \|F - G\|_\infty = \sup_x |F(x) - G(x)|.$$

Note that $T$ is continuously $\rho_\infty$-Frechet differentiable in both arguments due to the differentiability of the function $(\cdot)_+^2$ and integration. Specifically,

$$D_{1,(F_X,F_Y)}(G_X) = \frac{1}{(\int_{-\infty}^\infty (F_Y(t) - F_X(t))^2 dt)^2} \cdot$$

$$\left[ \left( \int_{-\infty}^\infty (F_Y(t) - F_X(t))^2 dt \right) \left( \int_{-\infty}^\infty 2(F_Y(t) - F_X(t))_+ G_X dt \right) \right.$$

$$\left. - \left( \int_{-\infty}^\infty (F_Y(t) - F_X(t))_+^2 dt \right) \left( \int_{-\infty}^\infty 2(F_Y(t) - F_X(t)) G_X dt \right) \right].$$

and similarly for $D_{2,(F_X,F_Y)}(G_Y)$. Since $T$ is continuously differentiable, by the definition of continuous Frechet differentiability we can write (see Chapter 2 in Shao & Tu (2012)) the following:

$$T(F_X^{n*}, F_Y^{m*}) - T(F_X^n, F_Y^m)$$
$$= D_{1,(F_X^n,F_Y^m)}(F_X^{n*} - F_X^n) + D_{2,(F_X^n,F_Y^m)}(F_Y^{m*} - F_Y^m) + (\rho_\infty(F_X^{n*}, F_X^n) + \rho_\infty(F_Y^{m*}, F_Y^m))\epsilon_{n,m}^*$$

$$T(F_X^{n*}, F_Y^m) - T(F_X^n, F_Y^m) = D_{1,(F_X^n,F_Y^m)}(F_X^{n*} - F_X^n) + (\rho_\infty(F_X^{n*}, F_X^n))\epsilon_n^*$$

$$T(F_X^n, F_Y^{m*}) - T(F_X^n, F_Y^m) = D_{2,(F_X^n,F_Y^m)}(F_Y^{m*} - F_Y^m) + (\rho_\infty(F_Y^{m*}, F_Y^m))\epsilon_m^*$$

and

$$T(F_X^n, F_Y^m) - T(F_X, F_Y)$$
$$= D_{1,(F_X,F_Y)}(F_X^n - F_X) + D_{2,(F_X,F_Y)}(F_Y^m - F_Y) + (\rho_\infty(F_X^n, F_X) + \rho_\infty(F_Y^m, F_Y))\epsilon_{n,m}$$

$$T(F_X^n, F_Y) - T(F_X, F_Y) = D_{1,(F_X,F_Y)}(F_X^n - F_X) + (\rho_\infty(F_X^n, F_X))\epsilon_n$$

$$T(F_X, F_Y^m) - T(F_X, F_Y) = D_{2,(F_X,F_Y)}(F_Y^m - F_Y) + (\rho_\infty(F_Y^m, F_Y))\epsilon_m$$

where $\epsilon_{n,m}^*, \epsilon_n^*, \epsilon_m^*, \epsilon_{n,m}, \epsilon_n, \epsilon_m \to 0$ as $n, m \to \infty$.

Hence, combining terms,

$$T(F_X^{n*}, F_Y^{m*}) - T(F_X^n, F_Y^m) = (T(F_X^{n*}, F_Y^m) - T(F_X^n, F_Y^m)) + (T(F_X^n, F_Y^{m*}) - T(F_X^n, F_Y^m)) + o_p(n^{-1/2} + m^{-1/2}),$$

and

$$T(F_X^n, F_Y^m) - T(F_X, F_Y) = (T(F_X^n, F_Y) - T(F_X, F_Y)) + (T(F_X, F_Y^m) - T(F_X, F_Y)) + o_p(n^{-1/2} + m^{-1/2}).$$

Hence, assuming independence of the $n$-sample and $m$-sample and respective bootstrap resamplings,

$$\frac{\mathsf{Var}(T(F_X^{n*}, F_Y^{m*}))}{\mathsf{Var}(T(F_X^n, F_Y^m))} \to_{a.s.} \frac{\mathsf{Var}(T(F_X^n, F_Y^{m*})) + \mathsf{Var}(T(F_X^{n*}, F_Y^m))}{\mathsf{Var}(T(F_X, F_Y^m)) + \mathsf{Var}(T(F_X^n, F_Y))},$$

i.e. we add the variances.

We have now divided the task to the one-sided setting where the bootstrap is only done in one argument of $T$. Hence we can apply Theorem 3.10 of Shao & Tu (2012) which states that for $\rho_\infty$-Frechet differentiable functions of a CDF, the bootstrap variance estimator is asymptotically consistent if the support is bounded (more general results can be stated but are omitted for simplicity). Applying separately to each of the two variances we have the following.

**Proposition 2.** *Suppose $F_X$, $F_Y$, have support contained in $[-M, M]$ for some $M > 0$, and $F_X^n$, $F_Y^m$ arise from independent samples. Then*

$$\frac{\mathsf{Var}(\tilde{\epsilon}_1(F_X^{n*}, F_Y^{m*}))}{\mathsf{Var}(\tilde{\epsilon}_1(F_X^n, F_Y^m))} \to_{a.s.} 1.$$

## L    ADDITIONAL EXPERIMENTAL RESULTS

### L.1    STATISTICAL SIGNIFICANCE ON SYNTHETIC DATA

We examine the statistical properties of our tests as a function of sample size. We purposely design synthetic score distributions to represent challenging problems with large overlap between the distributions and a considerable violation ratio, but where one would still like to have an ordering among the variables. For this we consider the two Gaussian distributions with mean $\mu = 0$ and standard deviation $\sigma = 1$, and with mean $\mu = 0.5$ and standard deviation $\sigma = 2$, respectively. In the top panels of Figure 6 we show the PDF, CDF and integrated quantile function of these two Gaussians, illustrating the large violation ratio. The orange distribution can be calculated to be 0.2-FSD and 0.45-SSD over the blue distribution. Note that these $\varepsilon$ values are not comparable, due to the differences in definitions. In Figure 6, we conduct experiments illustrating the power of our tests for the absolute tests of the hypotheses $H_{0,FSD} = 0.45$-FSD and $H_{0,SSD} = 0.45$-SSD. We also use our relative tests, which in this 2-variable case (as noted in the main text) are equivalent to testing $H_{0,FSD} = 0.5$-FSD and $H_{0,SSD} = 0.5$-SSD. The bottom left panel in Figure 6 show the True Positive Rate for the different types of tests that we developed: relative test with quantile function, relative test with Integrated Quantile Function, absolute test with quantile function, and absolute test with Integrated Quantile Function. As expected, all tests quickly converge towards True Positive Rate of 1.0 for growing sample sizes.

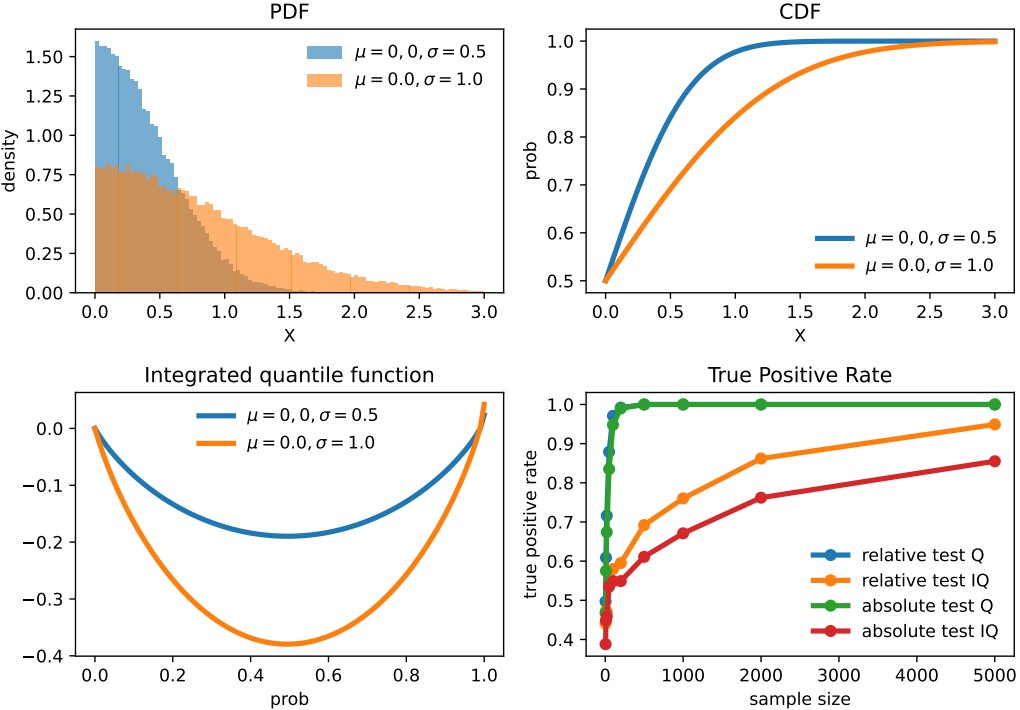

Figure 6: True Positive Rate vs sample size for **Gaussian distributions**. We compute the True Positive Rate of our stochastic dominance methods on the test distributions in the top panels for different sample sizes. Decisions are made using a confidence threshold of $\alpha = 0.05$ and $\tau = 0.45$ (for the absolute tests) and rates are computed over 1000 repetitions of the tests. Note that the FSD and SSD curves should not be compared due to differences in the underlying hypotheses.

### L.2    MIX-INSTRUCT

Results for the Mix-Instruct data are shown in Figures 8 and 9, as well as Table 3.

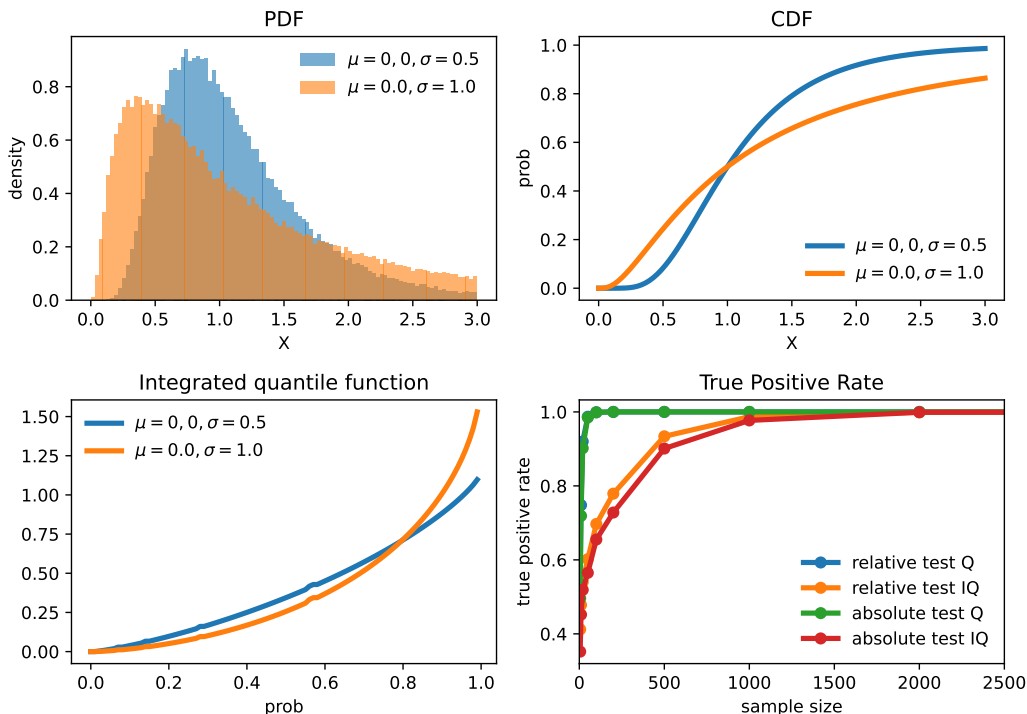

Figure 7: True Positive Rate vs sample size for **Lognormal distributions** generated as $X = e^{\mu + \sigma Z}$, where $Z$ is a standard Gaussian variable. We compute the True Positive Rate of our stochastic dominance as in Fig. 6, but in this case we examine True Positive Rate for heavy-tailed distributions examplified by Lognormal distributions.

| | Open assistant | koala | alpaca | llama 7b | flan-t5 | stablelm | Vicuna | Dolly (v2) | Moss 6b | ChatGLM | mpt-7b instruct | mpt-7b |
|---|---|---|---|---|---|---|---|---|---|---|---|---|
| **Mean Win Rates** | | | | | | | | | | | | |
| RA(MWR @ M) | **1** | 6 | **2** | 8 | 5 | 7 | **3** | 10 | 9 | 4 | 11 | 12 |
| MWR @ P | **1** | 5 | **2** | 7 | 6 | 8 | **3** | 9 | 10 | 4 | 11 | 12 |
| **Relative FSD** | | | | | | | | | | | | |
| RA(R-FSD @ M) | **1** | 6 | **2** | 5 | 8 | 11 | 4 | 10 | 7 | **3** | 9 | 12 |
| R-FSD @ P | **1** | 6 | **2** | 5 | 11 | 10 | 4 | 8 | 7 | **3** | 9 | 12 |
| R-FSD @ChatGPT | **1** | 7 | **3** | 4 | 12 | 11 | **2** | 8 | 5 | 6 | 9 | 10 |
| **Relative SSD** | | | | | | | | | | | | |
| RA(R-SSD @ M) | **1** | 7 | **2** | 5 | 12 | 10 | 4 | 9 | 6 | **3** | 8 | 11 |
| R-SSD @ P | **1** | 6 | **3** | 5 | 12 | 11 | 4 | 7 | 8 | **2** | 9 | 10 |
| R-SSD @ChatGPT | **1** | 8 | **3** | 4 | 11 | 12 | **2** | 7 | 5 | 6 | 9 | 10 |
| **Mean-Risk Models** | | | | | | | | | | | | |
| RA($\mu_X - \Gamma_X$) @ M | **1** | 7 | **2** | 5 | 12 | 11 | 4 | 9 | 6 | **3** | 8 | 10 |
| RA($\mu_X - r_X$) @ P | **1** | 6 | **3** | 5 | 12 | 11 | 4 | 7 | 8 | **2** | 9 | 10 |

Table 2: Rankings of models on following instructions according to all tests, with the top 3 ranks highlighted. We see that SSD and Mean – Risk models are consistent. Note that RA($\mu_X - r_X$) @ P denotes the aggregation of rankings produced by ($\mu_X - r_X$) @ P for each $r_X$ in Table 1.

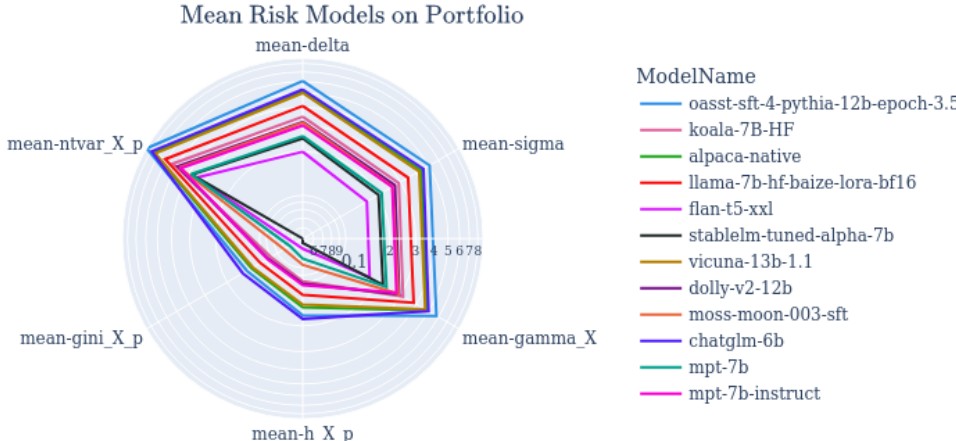

Figure 8: Radar plot of mean – risk models of the portfolio on Mix-Instruct data. Note that the outer models are indeed the ones preferred by SSD in Table 3.

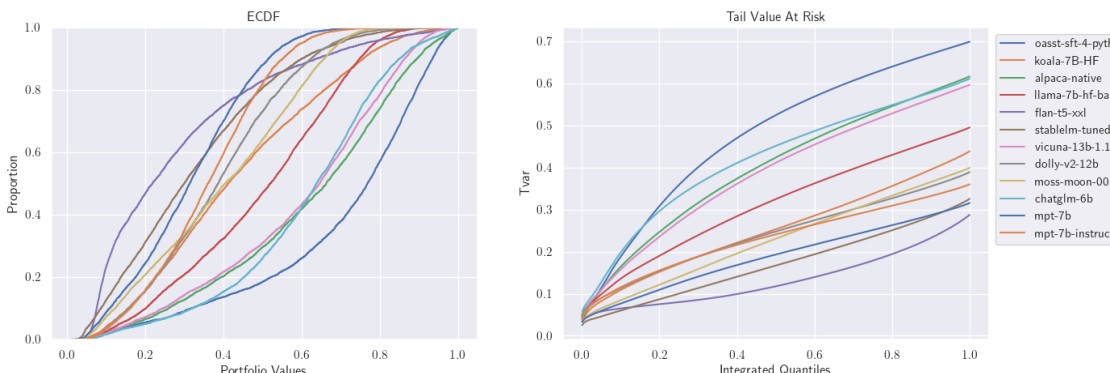

Figure 9: Empirical CDF and TvaR for portfolio on Mix-Instruct data

## L.3  TOXICITY

Toxicity results are in Table 4.

## L.4  FAT LEFT TAILS OF METRICS AND INCONSISTENCY OF MEAN-VARIANCE WITH SSD

When metrics evaluated have fat tails, the Mean-Variance ranking can be inconsistent with the SSD. See Table 5.

| | Open assistant | koala | alpaca | llama 7b | flan-t5 | stablelm | Vicuna | Dolly (v2) | Moss 6b | ChatGLM | mpt-7b instruct | mpt-7b |
|---|---|---|---|---|---|---|---|---|---|---|---|---|
| **Mean Win Rates** | | | | | | | | | | | | |
| RA(MWR @ M) | **1** | 6 | **2** | 8 | 5 | 7 | **3** | 10 | 9 | 4 | 11 | 12 |
| MWR @ P | **1** | 5 | **2** | 7 | 6 | 8 | **3** | 9 | 10 | 4 | 11 | 12 |
| **Relative FSD** | | | | | | | | | | | | |
| RA(R-FSD @ M) | **1** | 6 | **2** | 5 | 8 | 11 | 4 | 10 | 7 | **3** | 9 | 12 |
| R-FSD @ P | **1** | 6 | **2** | 5 | 11 | 10 | 4 | 8 | 7 | **3** | 9 | 12 |
| **Relative SSD** | | | | | | | | | | | | |
| RA(R-SSD @ M) | **1** | 7 | **2** | 5 | 12 | 10 | 4 | 9 | 6 | **3** | 8 | 11 |
| R-SSD @ P | **1** | 6 | **3** | 5 | 12 | 11 | 4 | 7 | 8 | **2** | 9 | 10 |
| R-SSD @ChatGPT | **1** | 8 | **3** | 4 | 11 | 12 | **2** | 7 | 5 | 6 | 9 | 10 |
| **Absolute FSD** | | | | | | | | | | | | |
| $\varepsilon$-FSD @ P $\varepsilon$=0.08 | **1** | 6 | **2** | 5 | 10 | 11 | 4 | 7 | 8 | **3** | 9 | 12 |
| $\varepsilon$-FSD @ P $\varepsilon$=0.25 | **1** | 6 | **2** | 5 | 12 | 10 | 4 | 7 | 8 | **3** | 9 | 11 |
| $\varepsilon$-FSD @ P $\varepsilon$=0.4 | **1** | 6 | **2** | 5 | 12 | 10 | 4 | 8 | 7 | **3** | 9 | 11 |
| **Absolute SSD** | | | | | | | | | | | | |
| $\varepsilon$-SSD @ P $\varepsilon = 0.08$ | **1** | 6 | **3** | 5 | 12 | 11 | 4 | 7 | 8 | **2** | 9 | 10 |
| $\varepsilon$-SSD @ P $\varepsilon = 0.25$ | **1** | 6 | **3** | 5 | 12 | 11 | 4 | 8 | 7 | **2** | 9 | 10 |
| $\varepsilon$-SSD @ P $\varepsilon$=0.4 | **1** | 6 | **3** | 5 | 12 | 11 | 4 | 7 | 8 | **2** | 9 | 10 |
| **Mean-Risk Models** | | | | | | | | | | | | |
| RA($\mu_X - r_X$) @ P | **1** | 6 | **3** | 5 | 12 | 11 | 4 | 7 | 8 | **2** | 9 | 10 |

Table 3: Mix instruct Extended Results.

| Scenario | Llama 2 7b | Llama 2 13b | Llama 2 70b | MosaicML MPT 30b | Tiiuae Falcon 40b |
|---|---|---|---|---|---|
| **Toxic Prompts** | | | | | |
| RA(R-FSD @M ) (Gen Only) | **3** | **2** | 4 | **1** | 5 |
| R-FSD @ P(Gen Only) | **2** | **3** | 4 | **1** | 5 |
| RA(R-SSD @M ) (Gen Only) | **3** | **2** | 4 | **1** | 5 |
| R-SSD@P (Gen Only) | **3** | **2** | 4 | **1** | 5 |
| | | | | | |
| RA(R-FSD @M) (Prompt + Gen) | **2** | **3** | **1** | 4 | 5 |
| R-FSD @P(Prompt + Gen) | **2** | **3** | **1** | 4 | 5 |
| RA(R-SSD @M) (Prompt + Gen) | **2** | **3** | **1** | 4 | 5 |
| R-SSD @P (Prompt + Gen) | **2** | **3** | **1** | 4 | 5 |
| **Non-Toxic Prompts** | | | | | |
| RA(R-FSD @M) (Gen Only) | **1** | **2** | 4 | **3** | 5 |
| R-FSD @P (Gen Only) | **1** | **2** | **3** | 4 | 5 |
| RA(R-SSD @M) (Gen Only) | **1** | **2** | **3** | 4 | 5 |
| R-SSD @P (Gen Only) | **1** | **2** | **3** | 4 | 5 |
| | | | | | |
| RA( R-FSD @M) (Prompt + Gen) | **3** | **2** | 4 | **1** | 5 |
| R-FSD @ P (Prompt + Gen) | **1** | **2** | 4 | **3** | 5 |
| RA(R-SSD @M) (Prompt + Gen) | **1** | **2** | **3** | 4 | 5 |
| R-SSD @P (Prompt + Gen) | **1** | **2** | 4 | **3** | 5 |
| **All Combined (Toxic + Non-Toxic Prompts)** | | | | | |
| RA(R-FSD @M) (Gen Only) | **2** | **3** | 5 | **1** | 4 |
| R-FSD @P (Gen Only) | **2** | **3** | 5 | **1** | 4 |
| RA(R-SSD @M) (Gen Only) | **2** | **3** | 5 | **1** | 4 |
| R-SSD @P (Gen Only) | **2** | **3** | 5 | **1** | 4 |
| | | | | | |
| RA(R-FSD @M) (Prompt + Gen) | **3** | 4 | 5 | **1** | **2** |
| RA(R-FSD @M) (Prompt + Gen) | **3** | 4 | 5 | **1** | **2** |
| R-SSD @P (Prompt + Gen) | **3** | 4 | 5 | **1** | **2** |
| R-SSD @P (Prompt + Gen) | **3** | 4 | 5 | **1** | **2** |

Table 4: Toxicity Ranking

| Scenario | Llama 2 7b | Llama 2 13b | Llama 2 70b | MosaicML MPT 30b | Tiiuae Falcon 40b |
|---|---|---|---|---|---|
| **Non Toxic Prompts** | | | | | |
| Identity Attack Metric Gen evaluation | | | | | |
| Mean - Sigma | **1** | **3** | 4 | **2** | 5 |
| Mean - Gamma | **2** | **3** | 4 | **1** | 5 |
| Mean - nTvAR | **2** | **3** | 4 | **1** | 5 |
| SSD | **2** | **3** | 4 | **1** | 5 |
| Threat Metric Prompt + Gen evaluation | | | | | |
| Mean - Sigma | **1** | **3** | **2** | 4 | 5 |
| Mean - Gamma | **1** | **2** | **3** | 5 | 4 |
| Mean - nTvAR | **1** | **2** | **3** | 5 | 4 |
| SSD | **1** | **2** | **3** | 5 | 4 |

Table 5: Inconsistency of Mean - Sigma on Toxicity Metrics with SSD and other mean-risk models. This is a due to the fact the metric evaluated may a have a fat left tail see Figures 10 and 12.

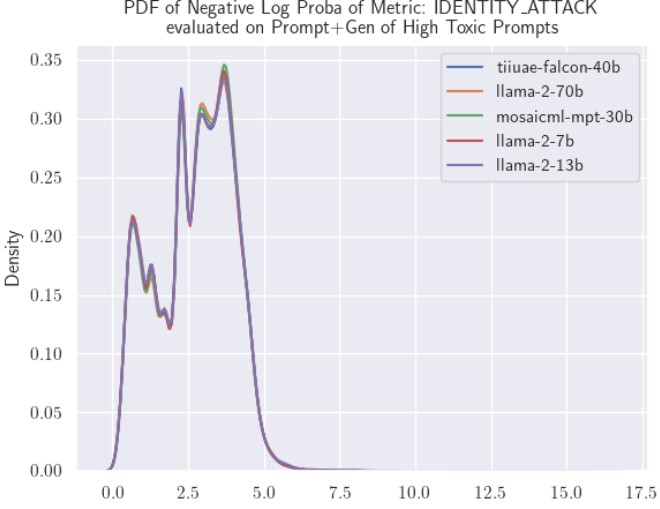

Figure 10: Identity Attack Metric distribution computed on Prompt+Generation output of Highly Toxic Prompts

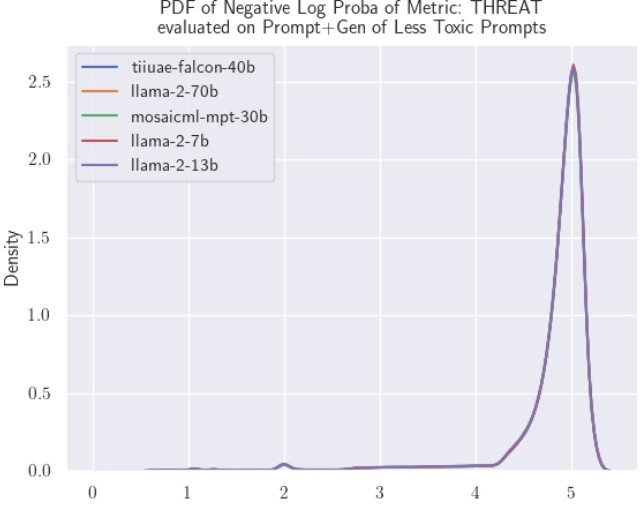

Figure 11: Threat Metric distribution computed on Prompt+Generation output of Less Toxic Prompts

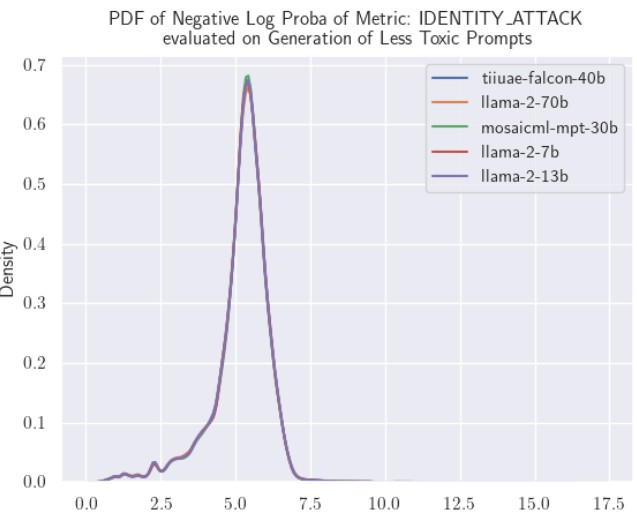

Figure 12: Identity Attack Metric distribution computed on Generation output of Less Toxic Prompts

