# OpenReview forum: "Risk Assessment and Statistical Significance in the Age of Foundation Models"
_ICLR.cc/2024/Conference — Submitted to ICLR 2024_

### Official Review · Reviewer_MwiU · 2023-10-30

**Soundness:** 2 fair
**Presentation:** 3 good
**Contribution:** 2 fair
**Rating:** 3
**Confidence:** 3

**Summary:**

This paper studies the assumption-less assessment of multiple models. The measures for comparison
are based on first and second-order stochastic dominance of real random variables. The authors further
provide an estimation scheme for the measures of interest, and complement the method with CLT-type theories.
The proposed method is used for comparing multiple large language models in several risk measures.

**Strengths:**

1. As promoted in the paper, with the rapid development of foundation models, there is a
lack of statistical significance testing---in this regard, the paper indeed investigates an interesting and important problem, and
provides a working solution.

2. I also find it valuable that the paper brings concepts from other fields, e.g., mathematical finance, to
task of model comparison.

**Weaknesses:**

1. I think more discussion on the motivation/interpretability of the proposed measures is needed. For example, why is R-SSD a relevant and interpretable measure for comparing two language models and why would metrics portfolio be a reasonable way to combine different metrics?

2. Although there are CLT results for the estimators, the validity of the statistical test is lacking --- in particular,
the validity of the Bootstrap approximation. The CLT result is not sufficient for the validity of the
test since the limiting distribution contains the unknown variance term; does Bootstrap approximation solve this problem? A theory is needed.
(Please correct me if I missed anything.)

**Questions:**

Please see the weaknesses part.

---

> ### Author Response · Authors · 2023-11-20
> **Response to Reviewer  MwiU**
>
> We thank the reviewer for their comment and address their questions:
>
> ___
>
> >**I think more discussion on the motivation/interpretability of the proposed measures is needed. For example, why is R-SSD a relevant and interpretable measure for comparing two language models and why would metrics portfolio be a reasonable way to combine different metrics?**
>
> We have added in the revised paper in the introduction more motivation and interpretability of the portfolio and the method  based on the SSD used in the paper.  Please see page 1-3 in the revision. The motivation for R-SSD is now depicted in Figure 1 contrasting panels (a) for quantiles (b) Tail Value at Risk that is normalized second quantiles . Second quantiles give better order between LLMs, since they assess left tails and failure modes of the models. Regarding why the portfolio reasonable, please see the points below:
> **Portfolio with R-SSD consistent with chatGPT score (proxy of human evaluation)**: We have incorporated in the revised manuscript comparison on the mix-instruct between a proxy of human evaluation given by chatGPT scores that evaluates LLMs and our portfolio approach. We have discussed this comparison in Figure 1 in the paper as well as in the introduction. We see our approach based on relative SSD is well aligned with chatGPT score, while the Min Win Rate (MWR) approach used widely in LLM  leaderboards such as HELM leads to very poor alignment with ChatGPT score.
> **Portfolio with R-SSD is consistent with rank aggregation of per metric rankings**:  In the revised manuscript, on the Mix-instruct dataset we show in Appendix A (page 12,        Metrics Aggregation Versus Portfolio) that Portfolio with R-SSD is consistent with rank aggregation of per metric rankings. Indeed these two approaches lead to similar ranking, which justify that portfolio is not losing the comparison power of models, but portfolio based is 7x faster than per metric based ranking (this speedup is taking into account the portfolio computation, note that in this example the number of metrics computed is 8).
>
> We hope this clears up the motivation and the interpretability of both SSD and portfolio.
>
> ___
>
> > **Although there are CLT results for the estimators, the validity of the statistical test is lacking --- in particular, the validity of the Bootstrap approximation. The CLT result is not sufficient for the validity of the test since the limiting distribution contains the unknown variance term; does Bootstrap approximation solve this problem? A theory is needed. (Please correct me if I missed anything.)**
>
>
> Thanks for this question. We have addressed this in page 7 in the manuscript and in Appendix K. One can compute the asymptotic variance of our tests using estimate of the density and the CDF but this is not a good option in finite sample regime, that is why we followed previous works such as  (Ulmer et al., 2022) that used a bootstrapping heuristic to perform testing of almost first order stochastic dominance.
> For the proof of consistency of bootstrapping, classical techniques of proofs require Hadamard or Frechet differentiability of the statistics. Note that our statistics are relaxations of FSD and SSD that can be either expressed in terms of CDFs and integrated CDF respectively or with quantiles; and integrated quantiles respectively. Appendix K considers statistics with CDFs and proves the consistency of the bootstrapping using classical results in (Shao & Tu, 2012). For the statistics with quantiles, a smoothing of the statistics is needed (Shao & Tu, 2012) for a similar proof to work, we leave this for future work since the length and detail of the derivation would be beyond the scope of the current work.

---

### Official Review · Reviewer_7hNz · 2023-10-31

**Soundness:** 3 good
**Presentation:** 3 good
**Contribution:** 3 good
**Rating:** 6
**Confidence:** 3

**Summary:**

The paper introduces a novel framework for assessing the socio-technical risks associated with foundation models, particularly large language models (LLMs), while providing a quantified measure of statistical significance. The framework relies on statistical tests based on first and second-order stochastic dominance, which are commonly used in economics and finance to balance risk and utility when making decisions.

The central idea is to evaluate LLMs across various metrics to assess risks, such as drifting from instructions and generating toxic content. The authors emphasize that these evaluations should encompass a wide range of tasks and domains. They cite several benchmark datasets and metrics, both automatic and human-based, commonly used for evaluating LLMs in various areas, such as chatbot performance and knowledge assessment.

The paper introduces second-order stochastic dominance (SSD) into the evaluation of LLMs, which the authors claim provides a more nuanced assessment of risk compared to traditional metrics like standard deviation.

The paper outlines the statistical tests for assessing almost and relative stochastic dominance and their theoretical underpinnings. They highlight the importance of central limit theorems and bootstrapping in estimating the statistical significance of these tests.

**Strengths:**

- The paper introduces a potential method for comparing LLMs, a relatively new and unsolved problem.
- The authors introduce a new asymptotic statistical test for a notion of stochastic ordering called relative stochastic dominance.

**Weaknesses:**

- The paper could be more clear on what it's main contribution is supposed to be. Is it "relative stochastic dominance" as a concept, the application of it to LLMs, the formulation of an asymptotic test statistic?

**Questions:**

- Is relative stochastic dominance stable with respect to the sets of stochastic variables involved? I.e. if I have two variables X_1 > X_2, and I add a new variable X_3, could I have X_3 > X_2 > X_1?
- Table 2 appears to suggest that mean-risk models give virtually equivalent rankings to relative SSD, yet are easier to understand. Why should one use relative SSD instead then?
- What aspect of your method is specific to LLMs as opposed to general model comparison along multiple dimensions?

---

> ### Author Response · Authors · 2023-11-20
> **Response to Reviewer 7hNz**
>
> We thank the reviewer for appreciating our work and their encouraging comments and suggestions. We address here their main questions:
>
> ___
>
> >**The paper could be more clear on what it's main contribution is supposed to be. Is it "relative stochastic dominance" as a concept, the application of it to LLMs, the formulation of an asymptotic test statistic?**
>
> We have improved in the revision the motivations for addressing the question of metric combination and statistical significance jointly, as well as the contributions. We also clarified the connection between the theory part that introduces the concepts and the application in evaluating LLMs. We have improved:
>
>   1. The presentation of the motivation and the contribution of our work in the introduction. Please see the edits in red in the revised manuscript in the introduction pages 1-3.
>  2. We gave more intuitions and introduced all concepts graphically (portfolio, FSD and SSD) in the introduction in **Figure 1** in a running example showcasing our contributions.
>  3. Throughout all theory sections we  kept reminding the reader in the theory section as suggested by the reviewer the role of each random variable as a metric evaluation or as a portfolio that is the aggregation of multiple metrics.  (Please see edits in red in the beginning of Section 2, we also simplified Section 3 and Section 4, without substantial change in the content).
>
> ___
>
> > **Is relative stochastic dominance stable with respect to the sets of stochastic variables involved? I.e. if I have two variables X_1 > X_2, and I add a new variable X_3, could I have X_3 > X_2 > X_1?**
>
> Let us assume that $X_1> X_2$ this means that $\varepsilon_1=\varepsilon_{12} < \varepsilon_2= \varepsilon_{21}$. When we add $X_3$, we have:
>
> $\varepsilon_1^{'}=\frac{1}{2}( \varepsilon_{12}+ \varepsilon_{13})$
> and
> $\varepsilon_2^{'}=\frac{1}{2}( \varepsilon_{21}+ \varepsilon_{23})$
> And
> $\varepsilon_3^{'}=\frac{1}{2}(\varepsilon_{31}+ \varepsilon_{32})$
>
> For $X_2$ to dominate $X_1$ in the new relative order considering 3 variables, we need to have $\varepsilon_2^{'} < \varepsilon_1^{'}  $, which means that we need to have: $\varepsilon_{23} - \varepsilon_{13}< \varepsilon_{12}- \varepsilon_{21} $ Note that the right side is negative by assumption.
>
> Hence the condition for the order to flip  between $X_1 $ and $X_2$ is:
>
>  $$\varepsilon_{23} - \varepsilon_{13}< \varepsilon_{12}- \varepsilon_{21} <0 $$
>
> Hence the order of variables of $X_1$ and $X_2$ can flip if  $\varepsilon_{23} < \varepsilon_{13}$, meaning that the violation of stochastic dominance of 2 on 3 is smaller than the one of 1 on 3 and if this delta  of violation is smaller than the delta of violation between 1 and 2.
>
> The order is only defined relative to the considered set. As we add variables, the order is sensible to the margins defined by the delta of violation ratios. That said, the flipping condition above should not occur in practice if the stochastic dominance between adjacent variables in the ordering is strong. In our own experiments, we find that the top-ranked models tend to be better-separated and more stable, the bottom-ranked models are often failing on many tasks and do not have a strong violation ratio between them, creating more unstable lower ranks.
>
> ___
>
>
> > **Table 2 appears to suggest that mean-risk models give virtually equivalent rankings to relative SSD, yet are easier to understand. Why should one use relative SSD instead then?**
>
> Mean Risk Models (MRM) are effectively useful, some of them are global statistics (integrated on all percentiles) like the Gini tail, but the tail value at risk, and the mean absolute  deviation   from a quantile are computed for a given percentile $p$ . We aggregate the ranks resulting from these MRM for a given $p$ using metric based rank aggregation (using pearson distance). Note that the resulting rank depends heavily on the percentile $p$ choice. The advantage of  Relative SSD testing, is that 1) statistically it has a confidence level which MRM lacks 2) Relative SSD is a global statistic that does not need any hyperparameter choice (except the confidence level). Hence Relative SSD is more reliable.
>
> To further compare Relative SSD and MRM we analyze the speed of their convergence versus sample size in Appendix A (Figure 2 page 12 in the revised paper), we see that R-SSD is more stable and has lower variance in low sample regime.
>
> **What aspect of your method is specific to LLMs as opposed to general model comparison along multiple dimensions?**
>
> The work motivation was LLM comparison, but as the reviewer is pointing this applicable to general model comparison along multiple dimensions, where these dimension measure risks, and in the LLM space lot of metrics are assessing those risks.

---

> > ### Comment · Reviewer_7hNz · 2023-11-22
> >
> > Thank you for the clarifications and improvements. I will maintain my score.

---

### Official Review · Reviewer_XDvM · 2023-11-07

**Soundness:** 1 poor
**Presentation:** 1 poor
**Contribution:** 2 fair
**Rating:** 5
**Confidence:** 4

**Summary:**

This paper proposed new evaluation metrics for comparing LLMs from two perspectives: (1) statistically robust risk assessment and (2) aggregation of multi-dimensional metrics. For the first part, the paper proposes R-SSD to estimate the difference of conditional value at risk (CVaR, Eq. (5)) confidently via relative pairwise testing. Then, for the second part, the paper proposes to use the weighted geometric mean as a "portfolio selection metric" inspired by portfolio optimization literature in finance. Finally, the paper conduct benchmark experiment of various LLMs using the proposed metric.

**Strengths:**

1. **The high-level motivation for doing a risk-aware evaluation of LLMs is well-described.** Evaluating the ability of LLMs is a very important topic nowadays, and having a risk-aware evaluation would indeed be useful.
2. **The paper benchmarks various SOTA LLMs.** The LLMs benchmarked by the proposed metrics includes various SOTA models, and doing a comprehensive benchmark experiment can be beneficial for the community.

**Weaknesses:**

1. **The main contribution of this paper is not very clear.** I have an impression that this paper discusses two independent topics in a paper. The first topic is the statistical robustness of risk assessment, and the second is how to balance tradeoffs among various metrics to compare LLMs. Though I admire that both of them have some motivation to work on, the issues of existing work in each topic are not clearly pointed out, and as discussed in the following Weakness 2, the paper fails to provide sufficient empirical analysis to compare the proposed method in both topics. Since the connection of two topics is also not very clear, I would suggest that splitting the paper into two might be useful.
2. **No quantitative or qualitative assessment criteria are given to demonstrate the effectiveness of the proposed evaluation method.** In my understanding, this paper proposes a new, statistically robust risk-assessment metric to compare two (multi-variate) probability distributions. However, the paper lacks an empirical demonstration of how the proposed method overcomes the issues of existing metrics. Given the focus of this paper, just applying the proposed metric to evaluate various LLMs is not enough. In addition, for the portfolio selection part, the paper should investigate if the proposed metric aligns with human preference more than other existing metrics to demonstrate the benefit of the proposed method.
3. **The manuscript is hard to follow.** The paper sometimes lacks explanations of theorems or motivations to introduce theorems. This makes understanding the main contribution of this paper quite challenging. In particular, the four points I mentioned in Questions were hard to understand from the manuscript.

**Questions:**

**Questions**
1. Isn't Eq. (4) the definition of conditional value at risk (CVaR)? (because it seems that (4) takes expectation of the quartile value under p% quartile). Then, deriving (4) seems straightforward, and I couldn't understand why the complex theory is needed for SSD. The manuscript should provide more explicit motivation for the theoretical analysis.

2. Are (1) and (2) always be equivalent? Similarly, are (3) and (5) always equivalent? In my understanding, (1) measures the cumulative probability of observing performance less than $\eta$, and (2) measures the $p$% quartile. It is not clear if (1) is always equivalent to (2) because they are measuring different quantities (though the statement may be true under some specific conditions about $\eta$ and $p$). Similarly, (3) measures the area under the CDF, while (5) measures CVaR. If the paper claims that (1) and (2) (or (3) and (5)) are equivalent, either citation or proofs should be included about this part.

3. What does the paper mean by "Note that FSD implies SSD, hence SSD is a finer notion of dominance"? Does it mean that FSD is a special case of SSD?

4. What is the intuitive understanding of $\epsilon$-FSD? Does it mean the probability that $X$ is preferred to $Y$ under the FSD criteria?

**Other minor things (typos)**
- page 4: period (.) is needed right before Eq. (7).
- page 8: 0,25 -> 0.25  (period (.) instead of comma (,))

---

> ### Author Response · Authors · 2023-11-20
> **Response to Reviewer XDvM**
>
> We thank the reviewer for their comments and suggestions and address here their main points:
>
> ___
>
> > **The main contribution of this paper is not very clear. I have an impression that this paper discusses two independent topics in a paper. The first topic is the statistical robustness of risk assessment, and the second is how to balance tradeoffs among various metrics to compare LLMs.**
>
> We have improved in the revision the motivations for addressing the question of metric combination and statistical significance jointly, as well as the contributions. We also clarified the connection between the theory part that introduces the concepts and the application in evaluating LLMs. We have improved:
>
>   1. The presentation of the motivation and the contribution of our work  in the introduction . Please see the edits in red in the revised manuscript in the introduction pages 1-3.
>  2. We gave more intuitions and introduced all concepts graphically (portfolio, FSD and SSD) in the introduction in **Figure 1** in a running example showcasing our contributions.
>  3. Throughout all theory sections we  kept reminding the reader in the theory section as suggested by the reviewer the role of each random variable as a metric evaluation or as a portfolio that is the aggregation of multiple metrics. (Please see edits in red in the beginning of Section 2, we also simplified Section 3 and Section 4, without substantial change in the content).
>
> ___
>
> > **No quantitative or qualitative assessment criteria are given to demonstrate the effectiveness of the proposed evaluation method. In addition, for the portfolio selection part, the paper should investigate if the proposed metric aligns with human preference more than other existing metrics to demonstrate the benefit of the proposed method.**
>
> We have incorporated in the revised manuscript comparison on the mix-instruct between a proxy of human evaluation given by chatGPT scores that evaluates LLMs and our portfolio approach. We have discussed this comparison in Figure 1 in the paper as well as in the introduction. We see our approach based on relative SSD is well aligned with chatGPT score, while the Min Win Rate (MWR) approach used widely in LLM  leaderboards such as HELM leads to very poor alignment with ChatGPT score.
> Furthemore, we give in appendix A Figure 3, a quantitative assessment of this alignment on per sample basis  between relative SSD test and chatGPT score , and  between MWR chatGPT score.  We see clearly that our approaches lead to a robust ranking that is consistent with chatGPT scores, while the MWR approach does not show any consistency with ChatGPT.
>
> ___
>
> > **The manuscript is hard to follow. The paper sometimes lacks explanations of theorems or motivations to introduce theorems. This makes understanding the main contribution of this paper quite challenging. In particular, the four points I mentioned in Questions were hard to understand from the manuscript.**
>
> We have taken into account all the mentioned points in our revision as clarified in our response to the first point in the rebuttal.
> ___
>
> > **Isn't Eq. (4) the definition of conditional value at risk (CVaR)? (because it seems that (4) takes expectation of the quartile value under p% quartile). Then, deriving (4) seems straightforward, and I couldn't understand why the complex theory is needed for SSD. The manuscript should provide more explicit motivation for the theoretical analysis.**
>
> Thanks for the suggestion. Yes this is indeed almost CVaR,  $CVaR(p)=\frac{ F^{(-2)}(p)}{p}$.  In the literature of risk analysis,  CVaR is also referred as Tail Value at Risk TVAR, we use in the paper TVAR to stress the tail analysis that SSD performs. We have now motivated SSD in the introduced from conditional value at risk point of view, as it can be seen graphically in panel b in figure 1. Thanks for the suggestion.
> ___
>
> > **(1) and (2) are equivalent ? (3) and (5) are equivalent ? what conditions ?**
>
> Yes (1) and (2) are equivalent;  (3) and (5)  are equivalent. We added in the paper reference to where proofs of these claims are given **without any assumptions needed** (Ogryczak & Ruszczynski, 2002).
> For (1) and (2), please note the CDF and quantiles are inverse one of another, hence if CDF of X is lower than CDF of Y , the quantiles will have the reverse relationship, Quantile of X are higher than Quantile of X.
>
> For (3) and (5), the second performance function (area under CDF), and the second quantile (area under quantiles )  have very interesting properties (( See Ogryczak & Ruszczynski, 2002 ):  They are convex dual one of another. Hence we can write :
> $$ F^{-2}(p) = \sup_{\eta} p\eta - F^{(2)}(\eta),$$
> Introducing the argument of the sup above in equation (3) and taking the sup on $\eta$ gives (5). A similar argument holds to go from (5) to (3).

---

> > ### Author Response · Authors · 2023-11-20
> > **Continued Response**
> >
> > > **What does the paper mean by "Note that FSD implies SSD, hence SSD is a finer notion of dominance"? Does it mean that FSD is a special case of SSD?**
> >
> > We meant that FSD implies SSD, i.e. if X dominates Y in FSD, we automatically have X dominates Y in SSD. But we can find random variables such that X > Y in SSD , but X < Y in FSD. In that sense SSD considers the tails of the distribution of the random variables and may be finer than FSD in a risk averse assessment of the order between the random variables. Given this interpretation, we would not use the terminology “special case.”
> >
> > ___
> >
> > > **What is the intuitive understanding of $\varepsilon-$ FSD?**
> >
> >  X> Y in $\varepsilon$ FSD means that all quantiles of X are higher than those of Y, except a smaller proportion of quantiles. This proportion where Y quantiles are higher, is the violation set.  $\varepsilon$ FSD  means that the area of this violation set is smaller than $\varepsilon$.

---

> ### Comment · Reviewer_XDvM · 2023-11-21
>
> Thank you for the detailed response and for uploading the revision. The additional comparison of the ChatGPT score would be helpful for the readers, and I appreciate the authors addressing some questions including the connection to CVaR.
>
> However, I still have remaining concerns and will keep my initial score. My concerns and suggestions are the following.
>
> ----
>
> - The paper is not self-contained. While I appreciate the authors' effort in providing additional explanations, I found that the paper refers to the appendix too many times when explaining the experimental results. Having further results in the appendix is not a negative thing, but even so, a paper should also be self-contained with its main text.
>
> - Even after reading the revision, I still have concerns about Weakness 1. I do understand that interpretability, risk assessment, and statistical significance can be several desiderata of evaluating LLM. However, I still do not understand why these independent topics should be discussed together in a paper. This is because there is no discussion on the tradeoff between these aspects, and independent discussions seem to easily fit with each other. Indeed, the paper merely combines the proposed approach on both, and the challenges in combining the discussion are not well-explained. The paper still lacks the motivation to work on several things altogether.
>
> - It seems imprecise to claim that "aggregated metrics are better for interpretability". Usually, decomposing a single metric into multiple aspects to explain the reason for a high score of a single metric is referred to as "interpretability" in machine learning. Such examples include SHAP [Lundburg & Lee '17]. The proposed aggregated metric seems to do the opposite thing. Associating the contribution of having aggregated metrics with "interpretability" sounds misleading and confusing.
>
> - While comparing the proposed metric and MWR against the ChatGPT score is helpful, the claim of Figure 1 (c) is not immediately clear. I suggest using quantitative comparisons, such as reporting rank correlation with the ChatGPT score.
>
> - (Also, the paper still needs improvements in the presentation, including the above points.)
>
> [Lundburg & Lee '17] Scott Lundberg, Su-In Lee. A Unified Approach to Interpreting Model Predictions. 2017. (NeurIPS)

---

> ### Author Response · Authors · 2023-11-21
> **Thank you for your response : re- interpretability and quantitative rank correlation**
>
> Thank you for your response!
>
> > **referring to Appendix**
>
> We have provided all ablations experiments and results requested by **6 reviewers** ,  given the space limitation, we had to refer to these in the appendix, and provide main experimental results in Figure 1.
>
> ___
>
> >**Indeed, the paper merely combines the proposed approach on both, and the challenges in combining the discussion are not well-explained. The paper still lacks the motivation to work on several things altogether**
> > **The proposed aggregated metric seems to do the opposite thing. Associating the contribution of having aggregated metrics with "interpretability" sounds misleading and confusing.**
> > **Usually, decomposing a single metric into multiple aspects to explain the reason for a high score of a single metric is referred to as "interpretability" in machine learning. Such examples include SHAP [Lundburg & Lee '17].**
>
> We respectfully disagree on this point, as we have explained the main challenges in multi-metric setting is that multiple metric can 1) have different polarity (meaning high is good or low is good) 2) and they can have also different dynamic range. This makes the interpretation of the metrics difficult , aggregation resolves this problem by bringing them to same polarity and same scale and returning a single  interpretable scalar. We have also confirmed that this aggregation to a single scalar is also in agreement with two baselines : chatGPT socre (human evaluation proxy) and per metric ranking aggregations, which validates the method.
>
> We believe that handling the three aspects of interpretability, statistical significance and risk assessment jointly is important. Since the communication of the results of such an interpretable scalar score with 1) statistical significance and 2) risk assessment will make it easier for a reviewer of an LLM or a regulator to interpret the results, especially when they lack the technical understanding of the metrics. Of course these aggregations can happen within dimensions as pointed by the reviewer.
>
> We also disagree respectfully that aggregation as a means of interpretability is misleading.  note that this aggregation of metrics for interpretability  is at the heart  of the index of transparency of foundation models introduce recently. The main motivation of this work is similar to us to come up with an interpretable score that easy to interpret and communicate of multiple metrics. This aggregation is a common practice in social sciences and is referred to as an index computation.
>
> The Foundation Model Transparency Index
> https://arxiv.org/abs/2310.12941
>
> ___
>
> > **While comparing the proposed metric and MWR against the ChatGPT score is helpful, the claim of Figure 1 (c) is not immediately clear. I suggest using quantitative comparisons, such as reporting rank correlation with the ChatGPT**
>
> We have provided quantitive comparison of rank correlation with kendall tau with the chatGPT score in **Appendix A Figure 3 (page 13)** the protfolio with risk assessment  leads to a kendall tau rank  similarity with chatGPT  of 0.75 while MWR with portfolio leads to a kendall tau rank  similarity with chatGPT  of 0.5.
>
>
> ___
>
> > **(Also, the paper still needs improvements in the presentation, including the above points.)**
>
> As we explained above regarding the presentation, the author seems concerned about  the motivation.  We believe the reviewer missed the important point that for any assessment framework , **the interpretability and the ease of communication** of what is being assessed, and its statistical significance comes hand in hand especially, when in it comes to socio-technical harms of LLMs  is of paramount importance. Aggregation method for interpretable communication are standard methods in social science when it comes to computing indices,  please see Stanford transparency index that applied this idea to LLMs recently.
>
> Regarding quantitative results we have already provided them.
>
> We hope this clarifies the reviewers questions!

---

> > ### Comment · Reviewer_XDvM · 2023-11-21
> >
> > Thank you for the response. Having read the rebuttals, I will maintain my evaluation based on the quality of the paper.

---

### Official Review · Reviewer_57J8 · 2023-11-07

**Soundness:** 3 good
**Presentation:** 3 good
**Contribution:** 3 good
**Rating:** 6
**Confidence:** 3

**Summary:**

This paper proposes a framework to evaluate and compare complex "foundation" models with quantified statistical significance. The proposed methodology has analogies with portfolio selection and optimization under risk aversion. First, an overview of the notion of stochastic dominance is performed, in terms of cumulative distribution and quantile functions. This involves covering relaxations of stochastic dominance and introducing the notion of relative dominance. Afterwards, asymptotically normal test statistics are derived to test for stochastic dominance and relative stochastic dominance. Finally, the presented methods are applied to assessing distributional risk in foundation models. The key idea here is to evaluate the model on multiple tasks that each output some sort of metric of interest, and then to pool together a "portfolio" of metrics, finally comparing the models by comparing their portfolios using the tests proposed in the paper. The method is then tested on some complex real-world models.

**Strengths:**

Nice and well-thought out application of formal statistical methodology to the context of complex generative models. I enjoyed reading the exposition on stochastic dominance, I found it clearly written and developed from scratch. The paper itself falls well into the general literature on using statistical testing to evaluate the output and/or performance of complex models. I also liked the idea of using a portfolio of metrics, this seems like a good way to aggregate the different metrics while being able to assign weights representing their importance to them.

**Weaknesses:**

For the novice, more intuition on the notion of *relative* stochastic dominance would be useful. Are there any conditions on defining the thresholds on the pairs of violations ratios, epsilon_ij?

I found the evaluation protocol and baselines section to be a bit too condensed. The way I am guessing the comparisons are done in the paper is to perform pairwise comparisons of all models, and then use the outcomes of the pairwise comparisons to produce a ranking?

**Questions:**

How does one choose the lambda_i's in (17)?

Is there a way to account for the possible correlations between the different metrics? The way the portfolio is constructed now does not seem to account for this, as it just takes a geometric average of the individual CDF's?

---

> ### Author Response · Authors · 2023-11-20
> **Response to Reviewer 57J8**
>
> Thanks a lot for your encouraging comments and for appreciating the work. We address here your questions:
>
> ___
>
> > **For the novice, more intuition on the notion of relative stochastic dominance would be useful.**
>
> Thanks for your comment, we have incorporated more intuition in the introduction describing all concepts graphically on a running example.
>
> We have improved in the revision the connection between the theory part that introduces the concepts and the application in evaluating LLMs. We have improved:
>   1. The presentation of the motivation and the contribution of our work in the introduction. Please see the edits in red in the revised manuscript in the introduction pages 1-3.
>  2. We introduced all concepts graphically (portfolio, FSD and SSD) in the introduction in **Figure 1** in a running example showcasing our contributions
>  3. Throughout all theory sections we keep reminding the reader (as suggested by the reviewer) the role of each random variable as a metric evaluation or as a portfolio aggregating of multiple metrics.  (Please see edits in red in the beginning of Section 2, we also simplified Section 3 and Section 4, without substantial change in the content).
>
> ___
>
> > **Are there any conditions on defining the thresholds on the pairs of violations ratios, epsilon_ij?**
>
> In our relative testing, after defining the all pairs  violation ratios $\varepsilon_{ij}$, we define a one versus all violation ratio $\varepsilon_i$ that is the average of all pairwise-ratios between model $i$ and all other models. This allows us to have a thresholdless approach to rank model, since now we simply test if $\varepsilon_i < \varepsilon_j$ to see model $i$ dominates model $j$. Note that this test does not need any threshold.
>
> In the absolute testing, the reviewer is right, it is not easy to select a single threshold $\varepsilon$ that is fair for all pairwise comparisons. This has motivated us to introduce the relative testing to remove the need for a threshold.
> ___
>
> > **I found the evaluation protocol and baselines section to be a bit too condensed. The way I am guessing the comparisons are done in the paper is to perform pairwise comparisons of all models, and then use the outcomes of the pairwise comparisons to produce a ranking?**
>
> Thanks for the comment, that is correct.  We have clarified this,  please see page 7 Multi-Testing Algorithm paragraph, and now Algorithm 1 Appendix C page 14  has clearer stepwise explanation of the Algorithm for how to go from all pairs comparisons to producing a single rank.
>
> ___
>
> >   **How does one choose the lambda_i's in (17)?**
>
> We used in portfolio uniform weights on the metrics. If there is a known order of importance between metrics, this order can be encoded in $\lambda$ and incorporated in the portfolio computation.
>
> ___
>
> >**Is there a way to account for the possible correlations between the different metrics? The way the portfolio is constructed now does not seem to account for this, as it just takes a geometric average of the individual CDF's?**
>
> Thank you for bringing up this important question. Without making assumptions on the metrics, it is not easy to incorporate the correlations. One possible idea would be to estimate vine copulas from the metrics, in a form of pairwise CDF between metrics and a graph of correlations (w) and define the portfolio as the vine copula $\exp\left(\sum_{ij} w_{ij} \log F(X_i, X_j)\right)$. This is an interesting question but is beyond the scope of the current work.

---

> > ### Comment · Reviewer_57J8 · 2023-11-22
> > **Response**
> >
> > Thank you to the authors for their response. I will retain my original score in this case, as I believe the aggregation technique can be refined. The copula idea is interesting, I agree.

---

### Official Review · Reviewer_wuox · 2023-11-08

**Soundness:** 2 fair
**Presentation:** 2 fair
**Contribution:** 2 fair
**Rating:** 5
**Confidence:** 2

**Summary:**

The paper deals with an interesting and relevant problem of risk assessment in foundation models. In particular, an approach from econometrics and finance is borrowed and extended to the problem at hand. This approach is based on the idea of mean-risk models, which are consistent with the second-degree stochastic dominance relation. The general idea behind mean-risk models is that there should be a balance between the expected return (mean) and associated risk. In the paper this idea is applied in the context of risk to generate toxic/harmful content.

Update after revision: I have increased my score after the reply from the authors. I believe the paper has improved, but the clarity is not sufficient yet for publication. Considering that I am not closely familiar with the portfolio risk evaluation methods presented in the paper, I do not feel I can judge the contribution with enough confidence. Hence, I also lowered my confidence score.

**Strengths:**

The general idea behind applying financial mathematics methods is interesting in the context of the problem addressed in the paper. With the rising number of various foundation models, it becomes increasingly important to construct robust evaluation approaches related to the risk of toxic/harmful content.

**Weaknesses:**

The connection between mathematical theory and the applied problem at hand is not emphasised enough. The transition between theoretical sections and experiments is somewhat abrupt. Simulation experiments validating statistical significance do not reflect real data experiments. Overall, I believe the paper can benefit from more clarity on the connection between the theory and the applied problem, a more realistic validation study on the simulated data and a much more explicit discussion of the failure modes of the methods (which assumptions need to be satisfied for the method to be theoretically justified and whether one indeed can check if these assumptions are satisfied in practice).

**Questions:**

-	In section 5.1 the approach – in particular, the validation of statistical significance – is validated on an example with two Gaussian random variables. I cannot quite connect this to the “more realistic” experiments later on. The fact that this works for two Gaussian random variables does not necessarily imply that it works in a similar way for more complex experiments.

-	The point above is also related to the underlying assumptions of the method. It is not uncommon to assume in some mean-risk models underlying Gaussian distribution. However, many attempts have been made to relax this assumption in finance since it is deemed to be unrealistic. How essential is it for the problem at hand?

-	What other assumptions need to hold for the method to have theoretical validity? How can one check if these assumptions indeed hold before applying the proposed method?

-	A lot of results are mixed and are presented in the appendix, even if described in the main text. I believe the paper could use restructuring to highlight the most relevant results and theory in the main text.

---

> ### Author Response · Authors · 2023-11-20
> **Response to Reviewer wuox**
>
> We thank the reviewer for their comments and address in the following their questions:
>
> ___
>
> > **The connection between mathematical theory and the applied problem at hand is not emphasized enough. The transition between theoretical sections and experiments is somewhat abrupt. Overall, I believe the paper can benefit from more clarity on the connection between the theory and the applied problem**
>
> Thanks for  your comment and suggestions.  We have improved in the revision the connection between the theory part that introduces the concepts and the application in evaluating LLMs. We have improved :
>   1. The presentation of the motivation and the contribution of our work  in the introduction . Please see the edits in red in the revised manuscript in the introduction pages 1-3.
>  2. We introduced all concepts graphically (portfolio, FSD and SSD) in the introduction in **Figure 1** in a running example showcasing our contributions
>  3. Throughout all theory sections we  kept reminding the reader in the theory section as suggested by the reviewer the role of each random variable as a metric evaluation or as portfolio that is the aggregation of multiple metrics.  (Please see edits in red in the beginning of Section 2, we also simplified Section 3 and Section 4, without substantial change in the content).
>
> ___
>
> > **In section 5.1 the approach – in particular, the validation of statistical significance – is validated on an example with two Gaussian random variables. I cannot quite connect this to the “more realistic” experiments later on. The fact that this works for two Gaussian random variables does not necessarily imply that it works in a similar way for more complex experiments.**
> > **The point above is also related to the underlying assumptions of the method. It is not uncommon to assume in some mean-risk models underlying Gaussian distribution. However, many attempts have been made to relax this assumption in**
> > **What other assumptions need to hold for the method to have theoretical validity? How can one check if these assumptions indeed hold before applying the proposed method?**
>
>
> We would like to clarify a misunderstanding of our work  that the reviewer seems to have. Our work is not about Mean Risk Models but on stochastic dominance. The Second Stochastic order is consistent with  many Mean Risk models. However, the mean-variance model that is used in finance is indeed problematic as it is not consistent with the Second order and can only be used if the two random variables compared are Gaussians.  The reviewer’s question is valid if we were using mean-variance models since they don’t apply beyond Gaussians. We are not indeed using these models.
>
> For all our stochastic dominance we need minor assumptions on the random variables , we collected these assumptions in page 6 (Assumption 1). The random variables must be bounded (or with bounded second moments), and have some regularity on their density.
>
> To illustrate that our stochastic dominance  tests are valid beyond Gaussianity on synthetic examples, we added in Appendix L.1 , our Relative First and Second order tests between  log normals. We choose log normal to illustrate two distributions that are heavy tails, which occurs a lot in practice.
>
> ___
>
> > **A lot of results are mixed and are presented in the appendix, even if described in the main text. I believe the paper could use restructuring to highlight the most relevant results and theory in the main text.**
>
> We hope the revision highlighted the relevant results and gave better intuitions.

---

### Official Review · Reviewer_qkuw · 2023-11-09

**Soundness:** 3 good
**Presentation:** 3 good
**Contribution:** 3 good
**Rating:** 6
**Confidence:** 3

**Summary:**

In a context where the random variables X and Y stand for the performance of two different models A and B respectively on a given metric, second order stochastic dominance (SSD) of X on Y implies that model A would be preferred over B by a risk-averse agent; i.e. one that prefers the lower-variability outcome given same expected performance/utility. Observing the desirability of such a comparison for evaluating machine learning models (esp. those likely to have large exposure to end users, such as language models), the current paper improves upon previous results from the econometrics literature to develop a statistical significance test for almost second order stochastic dominance (SSD) between random variables, as well as a variant of this test for comparing multiple random variables. They combine this test with a simple rank aggregation scheme, and a weighted geometric mean of different metrics (called metrics portfolio) to produce a ranking among a number of models, tested on the said metrics. They compare their method to other methods for ranking language models.

**Strengths:**

- The paper's topic of choice is timely: expanding on the available toolbox for quantifying and/or comparing the risks of multiple machine learning models, evaluated on multiple metrics is important in a context where foundation models are deployed in an unprecedented speed in various domains.
- The paper aptly utilizes previous results from econometrics literature, and expands succinctly on the said results when necessary, to produce statistical significance tests for SSD, which allows taking the variability of the model's performance into account in addition to expected performance when comparing.
- The writing is usually clear and easy to follow, and the paper concisely communicates various decision theoretic notions that are central to the present framework.

**Weaknesses:**

- How well the technical details of the paper's proposed methodology are communicated varies throughout the paper. Some central details are hard to decipher, in a way that hinders a complete understanding of the methodology. Please see the comments in the next section.
- There are long stretches where the text reads like it is from an econometrics paper, with the lack of emphasis on how the introduced concepts will be relevant for model comparison distracting from the main contributions of the paper. I recommend being more sensitive to the conference audiences' background, and frequently reminding the reader how a concept or result will help resolve an important sub/problem in the overall model comparison. See the first sentence of my summary of the paper for a simple example of an arguably more accessible framing.
- In the experiments, the paper's results with SSD almost perfectly line up with those obtained by using previously known mean-risk models: I believe that how exactly the current methodology improves upon these simple and useful models is not sufficiently discussed.

**Questions:**

- How the existing tests are applied pairwise and then combined to achieve multi-testing should be described and discussed in more detail (this applies to main paper and Appendix A.) I recommend:
  - Expanding the end of Section 3 to include further details of how the pairwise statistical tests are combined to produce the full ranking with a confidence of $1-\alpha$.
  - Expanding Appendix A to include a high-level description of Algorithm 1.
- What are the justifications for portfolio-based vs. per metric ranking of models?
- Why is $\lambda$ is not used (instead of uniform weighting) for rank aggregation between metrics in the latter method?
- What multi-way aggregation method is used after per metric ranking?
- How should $\epsilon$ be chosen? Why?
- Could the authors' methodology be extended to apply to metrics that are not continuous? (e.g. binary accuracy, human annotator ranking)

---

**Comments following the rebuttal period**: I thank the authors for their thorough response to reviewers' comments. Although some valid points are raised in terms of the limitations of the work, I believe the authors' responses and modifications they make to the paper are satisfactory. I retain my recommendation for acceptance.

---

> ### Author Response · Authors · 2023-11-20
> **Response to Reviewer qkuw**
>
> We thank the reviewer for appreciating the contributions of the paper, their questions suggestions that have greatly improved the paper. We address here their suggestions and questions:
> ___
>
>  > **Some central details are hard to decipher, in a way that hinders a complete understanding of the methodology. Please see the comments in the next section.**
> > **I recommend being more sensitive to the conference audiences' background, and frequently reminding the reader how a concept or result will help resolve an important sub/problem in the overall model comparison. See the first sentence of my summary of the paper for a simple example of an arguably more accessible framing.**
>
> Thanks for the suggestions, we have improved in the revision the connection between the theory part that introduces the concepts and the application in evaluating LLMs. We have improved :
>   1. The presentation of the motivation and the contribution of our work  in the introduction . Please see the edits in red in the revised manuscript in the introduction pages 1-3.
>  2. We introduced all concepts graphically (portfolio, FSD and SSD) in the introduction in **Figure 1** in a running example showcasing our contributions
>  3. Throughout all theory sections we  kept reminding the reader in the theory section as suggested by the reviewer the role of each random variable as a metric evaluation or as portfolio that is the aggregation of multiple metrics.  (Please see edits in red in beginning of Section 2, we also simplified Section 3 and Section 4, without substantial change in the content).
> ___
> > **In the experiments, the paper's results with SSD almost perfectly line up with those obtained by using previously known mean-risk models: I believe that how exactly the current methodology improves upon these simple and useful models is not sufficiently discussed.**
>
> Mean Risk Models (MRM) are effectively useful, some of them are global statistics (integrated on all percentiles) like the Gini tail, but the tail value at risk, and the mean absolute  deviation   from a quantile are computed for a given percentile $p$ . We aggregate the ranks resulting from these MRM for a given $p$ using metric based rank aggregation (using pearson distance). Note that the resulting rank depends heavily on the percentile $p$ choice. The advantage of  Relative SSD testing is that 1) statistically it has a confidence level which MRM lacks 2) Relative SSD is a global statistic that does not need any hyperparameter choice (except the confidence level ). Hence Relative SSD is more reliable.
>
> To further compare Relative SSD and MRM we analyze the speed of their convergence versus sample size in Appendix A ( Figure 2 page 12 in the revised paper), we see that R-SSD is more stable and has lower variance in low sample regime.
> ___
>
> > **Multi-testing should be described and discussed in more detail (this applies to main paper and Appendix A.)  I recommend ...of Algorithm 1.**
>
> Thanks for the suggestions, we have implemented them in main and in Appendix. (See page 7 Multi-Testing Algorithm paragraph, and now Algorithm 1 Appendix C page 14 has clearer explanation of the Algorithm )
>
> ___
>
> > **What are the justifications for portfolio-based vs. per metric ranking of models?**
>
> Thanks for the question. Portfolio based is computationally more efficient than per metric ranking of models. For per metric ranking, one needs to compute Relalive FSD or SSD for each metric, then use rank aggregation to obtain a ranking.  While for Portfolio based , one computes a single R-FSD or R-SSD and obtains the ranking of the models.
>
> In the revised manuscript, on the Mix-instruct dataset we show in Appendix A (page 12, Metrics Aggregation Versus Portfolio ) that indeed these two approaches lead to similar ranking, which justify that portfolio is not loosing the comparison power of models, but portfolio based is 7x faster than per metric based ranking (this speedup is taking into account the portfolio computation, note that in this example the number of metrics computed is 8).
>
> ___
> > **Why is $\lambda$ is not used (instead of uniform weighting) for rank aggregation between metrics in the latter method?**
>
> We used in portfolio and in rank aggregation uniform weights on the metrics. If there is a known order of importance between metrics , this order can be encoded in $\lambda$ and incorporated in both portfolio and rank aggregation of metric based ranking.
>
> ___
>
> > **What multi-way aggregation method is used after per metric ranking?**
>
> We discuss this in Appendix F.3 page 17 in revised manuscript,  we used metric based rank aggregation , with `pearson` as a distance.

---

> > ### Author Response · Authors · 2023-11-20
> > **continued response**
> >
> > >**How should $\varepsilon$ be chosen? Why?**
> >
> > It is not straightforward how to select $\varepsilon$, especially in multi-testing settings.  That is why we introduced the relative stochastic that is thresholdless. We recommend the use of our relative tests over the absolute test that require $\varepsilon$.
> > ___
> >
> > > **Could the authors' methodology be extended to apply to metrics that are not continuous? (e.g. binary accuracy, human annotator ranking)**
> >
> > Thanks, that is a very interesting question. Our test statistics are certainly defined for discrete valued metrics. Unfortunately the central limit theory in the paper relies heavily on the continuity of the random variables. Nevertheless, we can map human rankings  of multiple models to real-valued scores where our approach applies! We used these transformations on the chatGPT scores that we included in the revision; we followed Jiang et al. (2023) approach for this mapping. Please See appendix B page 13 that explains this approach. In future work, it would be interesting to explore the asymptotic distribution of the test statistics for discrete-valued metrics in order to design a valid hypothesis testing procedure in that setting - without needing to make the metrics continuous as we currently do.

---

### Comment · Area_Chair_HTu6 · 2023-11-10
**Authors-Reviewers discussion starts today, ends on Nov 22**

Dear authors and reviewers,

@Authors: please make sure you make the most of this phase, as you have the opportunity to clarify any misunderstanding from reviewers on your work. Please write rebuttals to reviews where appropriate, and the earlier the better as the current phase ends on Nov 22, so you might want to leave a few days to reviewers to acknowledge your rebuttal. After this date, you will no longer be able to engage with reviewers. I will lead a discussion with reviewers to reach a consensus decision and make a recommendation for your submission.

@Reviewers: please make sure you read other reviews, and the authors' rebuttals when they write one. Please update your reviews where appropriate, and explain so to authors if you decide to change your score (positively or negatively). Please do your best to engage with authors during this critical phase of the reviewing process.

This phase ends on November 22nd.

Your AC

---

### Author Response · Authors · 2023-11-20
**Comment to AC and all Reviewers**

Dear AC and Reviewers,

We thank the AC for overseeing this discussion and the reviewers for their feedback and suggestions that greatly improved our paper. We have uploaded a revision of the paper, edits are in red in the pdf.

1. We revised the introduction (pages 1-3), with more motivation of the work and explanation of the contributions on a running example conveying more intuitions on all the concepts introduced in the paper on both portfolio aggregation and stochastic dominance. As suggested by the reviewers, we also smoothed the transition between theory and application, so the reader is always reminded of the application of the theory.

2. We simplified the theory presentation and added all needed references for **equivalence between CDF and quantile approaches** (pages 4-5) and  **assumptions** for CLT to hold (page 6 Assumption 1)

3. We discuss **bootstrapping** in page  7 of quantile based statistics and added in Appendix K (page 25-26) a proof of bootstrap consistency for CDF based statistics relaxations of the stochastic dominance.

4. We clarified **Algorithm 1** in page 7 as a pairwise comparison with stochastic dominance, followed by finding ranking with the Borda rank aggregation of pairwise ranks; We also clarified the correction of the confidence level of this multi-testing setup. We added further explanation in Appendix B (page 13).

5. We added synthetic experiments for validating our statistical tests, on fat tailed distribution (log normal distribution, Figure 7 Appendix L.1 in page 28).

6. We added **validation of the portfolio** approach with **proxy of human evaluation** with chatGPT scores (See Appendix B page 13-14 for definition of this scoring) of the models on mix-instruct dataset (Figure 1, Table 1, see introduction and experiments page 9.

7. We clarified the advantage of **R-SSD on portfolio versus rank aggregation of R-SSD per metric**  as it leads to consistent ranks but it leads to almost linear speedups (in number of metrics).

8. We added a few **ablation studies** requested by the reviewers that can be found in Appendix A page 12-13. These ablations clarify that R-SSD has more stability in sample size than Mean Risk Models (MRM), thanks to its bootstrapping. We also clarified that some of these MRM are sensitive to the choice of the percentiles, whereas R-SSD are global statistics.

We hope that these extensive improvements address the concerns of the reviewers and improve their assessment of the paper. Please reach out during the remaining discussion period if you have further concerns!

---

### Meta-Review · Area_Chair_HTu6 · 2023-12-05

**Metareview:**

This meta-review is a reflection of the reviews, rebuttals, discussions with reviewers and/or authors, and calibration with my senior area chair. This paper investigates a risk assessment framework for foundation models. There is a consensus among reviewers that the paper has merits and presents interesting ideas, but unfortunately in a presentation which is hard to decipher at times, and there is a sentiment that the paper is unlikely to have a large impact unless significantly revised. I appreciate this will come as a disappointment to the authors: ICLR is a highly competitive venue and unfortunately the clarity of this submission is sup-optimal.

**Justification For Why Not Higher Score:**

Paper has merits but reviewers felt it is pretty hard to decipher in its current form, and I don't think it is likely to have a large impact unless significantly revised.

**Justification For Why Not Lower Score:**

N/A

---

### Decision · Program_Chairs · 2024-01-16

Reject